# Multi-Level Spatial Embedding Sharing for Enhanced Online Trajectory-User Linking

## Abstract

Trajectory-User Linking (TUL) is a critical task in mobility applications that links unlabeled spatial trajectories to the users or entities that generated them. In these applications, data often arrives as a continuous stream and may experience distributional shifts over time. While adapting TUL models via online learning could address these challenges, this approach remains unexplored in current research. Our work bridges this gap by conducting comprehensive evaluations of common TUL techniques in an online learning context. To improve the performance of existing TUL techniques in this setting, we propose Multi-Level Spatial Embedding Sharing (MiLES), an embedding approach that adapts and extends the principle of multi-scale spatial sharing for online TUL. MiLES partially shares embeddings across neighborhoods of multiple size levels, enabling generalization within neighborhoods while maintaining fine-grained discrimination through more location-specific representations. MiLES also reduces the number of embedding parameters by up to 38%, leading to lower memory usage and more computationally efficient model updates. We further incorporate a learnable scalar weight per embedding level that the model tunes during training to improve optimization in the sparse online setting. Our experimental results on several real-world datasets show that integrating MiLES into state-of-the-art TUL models improves their performance in online learning scenarios, yielding relative top-1 accuracy gains of roughly 2 to 5% over the standard POI lookup embedding on the POI-rich Foursquare datasets, and larger gains on the POI-free GeoLife dataset. On GeoLife, however, a continuous coordinate encoder performs better still, as such encoders appear better suited to its denser, continuous GPS data. Consistent improvements are also observed across other training paradigms. However, the online gains are particularly relevant, as our findings suggest that online learning is the most suitable paradigm real-time TUL on streaming data, outperforming periodic batch retraining at lower computational cost. To demonstrate its general applicability, we also evaluate MiLES on the task of destination prediction, where it provides consistent performance improvements, confirming its value as a domain-general embedding technique. Our code is available at `https://anonymous.4open.science/r/MiLES-3D20`.

## 1 Introduction

The proliferation of location-enabled devices has produced vast amounts of mobility data, fueling advances in various machine learning tasks (Rehman et al., 2015; Zheng, 2015). One such task is Trajectory-User Linking (TUL), introduced by Gao et al. (2017). TUL aims to associate chronologically ordered sequences of check-ins at visited locations or points of interest (POIs), so-called trajectories, to the users or entities that generated them. The TUL task has numerous real-world applications, including disease control, law enforcement, ride-sharing, and location-based recommendations (Hao et al., 2020; Gao et al., 2017). Traditional approaches for TUL have demonstrated impressive performance using recurrent neural networks and, more recently, transformer-based architectures.

However, these methods are designed for batch learning, where all training data is available beforehand, and to the best of our knowledge, no prior work has investigated TUL in online learning settings. In many practical scenarios, such as the deployment of a new location-based service, initial data is often limited, and

data instead arrives as a continuous stream. In addition, mobility data streams are subject to temporal dependencies and distributional shifts, commonly known as concept drift, caused by factors such as evolving user behavior, the emergence of new POIs, or external events like the COVID-19 pandemic (Borkowski et al., 2021). These challenges demonstrate the necessity for models that can adapt dynamically, which online learning addresses by allowing models to incrementally update their knowledge with new trajectory data. In contrast to traditional batch learning, online models process data sequentially as it arrives, enabling adaptation to changing mobility patterns. While periodic batch retraining is a common alternative, it requires retaining historical data, introduces latency between distribution changes and model adaptation, and incurs growing computational cost as the dataset accumulates. We show in Section 6 that online learning outperforms periodic retraining on all datasets while requiring less computation. Examples of further advantages include privacy-sensitive applications, where discarding trajectory data immediately after each model update reduces the risk of data breaches, and on-device deployment, where hardware constraints make full retraining costly.

According to the definition of online machine learning by Bifet et al. (2010), a model operating in such an environment must be able to:

**R1**: process a single instance at a time,
**R2**: process each instance in a limited amount of time,
**R3**: use a limited amount of memory,
**R4**: predict at any time,
**R5**: adapt to changes in the data distribution.

Most existing TUL approaches can be adapted for online TUL, which we formalize in Equation 2, by updating models incrementally. However, their ability to predict at any time and adapt to distributional shifts can be hindered by their embedding strategy. To embed check-in locations, most models rely on lookup tables containing separate learnable embedding vectors for each POI. For any given trajectory, only the embeddings for the visited locations receive non-zero gradients with this approach. However, given the large number of unique POIs in most TUL applications, most locations receive relatively few check-ins (Chen et al., 2022). This results in the gradients for POI-based embeddings often suffering from a high degree of sparsity. While in batch learning, this sparsity can be partially mitigated by training on the same trajectories for multiple epochs, it remains a limiting factor, particularly in online learning settings where each trajectory is observed only once and quick adaptation to new data is crucial (**R5**).

A potential solution is to share embeddings across multiple locations to reduce gradient sparsity, as was explored in previous works on batch-learning TUL (Yang et al., 2021; Alsaeed et al., 2023). Sharing embeddings based on spatial proximity allows models to generalize knowledge across locations, improving adaptation for rarely visited POIs. However, there is a trade-off: larger neighborhood sizes reduce the gradient sparsity, but also reduce the specificity of embeddings. This means that while broad embedding sharing may be beneficial at the start of training or following a concept drift, it likely degrades the model's ability to learn fine-grained location details.

To address this issue, we propose Multi-Level Spatial Embedding Sharing (MiLES), an embedding approach that adapts and extends the principle of multi-scale spatial representations (Mai et al., 2019; Hu et al., 2022). MiLES partitions each check-in's embedding vector into multiple parts, each shared across neighborhoods of a different size. This design enables a wide spectrum of representations with varying levels of specificity and gradient density, while also reducing the overall parameter count and improving the computational efficiency of model updates compared to a purely POI-based approach without shared embeddings. MiLES additionally features a learnable per-level weighting mechanism that adjusts the scale of each embedding level throughout the online learning process, which we find improves performance in the sparse data stream setting we study. While MiLES also improves performance under other training paradigms, we show in Section 6 that its benefits are particularly relevant in the online regime, where the model must generalize beyond its current training data.

To evaluate the effectiveness of MiLES, we first conduct a comprehensive assessment of existing state-of-the-art TUL techniques in an online learning setting. We then integrate MiLES into these techniques and

demonstrate its consistent performance improvements. Lastly, we perform ablation studies to analyze the contribution of each MiLES component.

While our primary focus is on TUL, the modular design of MiLES makes it applicable to other spatial machine learning tasks that benefit from multi-scale representations. To demonstrate this versatility, we successfully apply MiLES to destination prediction, showing its broader utility (see Section 7).

In summary, our main contributions are: (i) the first systematic evaluation of state-of-the-art TUL models in an online learning setting; (ii) MiLES, a multi-level spatial embedding sharing approach that adapts grid-based spatial sharing to the online TUL setting, reducing gradient sparsity through structured partitioning and learnable level weighting; (iii) an empirical demonstration that MiLES is an effective alternative to the standard POI-only lookup embedding used by most TUL models, improving top-1 accuracy across backbones by roughly 2 to 5% on Foursquare and by larger margins on GeoLife while reducing parameter counts, though on the latter's denser, continuous-GPS data a continuous coordinate encoder is more effective still; and (iv) a demonstration of MiLES's generality via the destination prediction task.

## 2 Preliminaries

In the following we will introduce the basic concepts underlying online trajectory user linking:

**Definition 2.1** (Check-In). A check-in is a tuple $\boldsymbol{c} = (t, \boldsymbol{l})$ containing a timestamp $t$ and the coordinates $\boldsymbol{l}$ of a visited location of form (latitude, longitude). Check-ins often additionally contain a unique identifier $p \in \mathbb{P}$ for the Point Of Interest (POI) at $\boldsymbol{l}$, where $\mathbb{P}$ is a finite set of POI identifiers.

**Definition 2.2** (Trajectory). A trajectory $T$ is a chronologically ordered sequence of $n$ check-ins, $T = [\boldsymbol{c}_0, \boldsymbol{c}_1, \ldots, \boldsymbol{c}_n]$. Each trajectory is generated by a single user $u$ from a set of users $\mathbb{U}$. A pair $(T, u)$ is a *linked trajectory*. When the generating user is unknown, the trajectory $T$ is considered *unlinked*.

**Definition 2.3** (Trajectory User Linking). The task of trajectory-user linking is to learn a function $f$ on a training set of linked trajectories $\mathbb{D} = \{(T_0, u_0), \ldots, (T_n, u_n)\}$, that correctly assigns a user label to each unlinked trajectory in a test set $\mathbb{D}^{(\text{test})}$. Formally, the objective is to minimize the predictive loss:

$$\min_{\boldsymbol{\theta}} \sum_{(T_i, u_i) \in \mathbb{D}^{(\text{test})}} L(f(T_i; \boldsymbol{\theta}, \mathbb{D}^{(\text{train})}), u_i), \tag{1}$$

where $L$ is a loss function quantifying the predictive error, and $\boldsymbol{\theta}$ are the model parameters to be optimized. In the batch setting, this objective reflects training on $\mathbb{D}^{(\text{train})}$ and evaluating on a held-out test set.

In an online setting, where samples arrive sequentially and cannot be stored indefinitely, TUL is more effectively evaluated under the *prequential* or *interleaved test-then-train* scheme (Bifet et al., 2010), where each incoming sample is first used to test the model and then for updating it. Given a stream of pairs $\mathbb{S} = \{(T_0, u_0), \ldots, (T_m, u_m)\}$, the objective becomes:

$$\min_{\boldsymbol{\theta}_0, \ldots, \boldsymbol{\theta}_{m-1}} \sum_{i=1}^{m-1} L(f(T_i; \boldsymbol{\theta}_{i-1}, \mathbb{S}_{:i-1}), u_i), \tag{2}$$

where $\boldsymbol{\theta}_{i-1}$ are the model parameters at time $i - 1$, and $\mathbb{S}_{:i-1}$ denotes all previously seen pairs. Therefore, the parameters at each step of the training process contribute equally to the performance of an online TUL model, making it more susceptible to the embedding sparsity issue mentioned in section 1.

Following the standard supervised online learning protocol (Bifet et al., 2010), we assume that the true user label $u_i$ becomes available after each prediction, allowing the model to update on $(T_i, u_i)$ before processing the next instance. This assumption is common in the online learning literature and is reasonable in applications such as ride-hailing, where the system predicts which user is most likely to service a requested route and observes the true assignment after completion. We acknowledge that settings with delayed or incomplete feedback may be more realistic for certain applications, and consider the extension of our approach to such settings a promising direction for future work.

## 3    Related Work

While trajectory user linking itself is a relatively recent task, it builds upon established methods from adjacent fields. Early approaches adapted the longest common subsequence (LCS) algorithm (Ying et al., 2011) to predict user labels by finding the longest shared sub-trajectory between unlabeled and known trajectories. Similarly, bag-of-words representations, which encode trajectories based on POI visit frequencies (Mikolov et al., 2013), enable the application of conventional classification methods such as linear discriminant analysis and support vector machines to the TUL problem.

More recent approaches, starting with TUL via Embedding and RNN (TULER) (Gao et al., 2017), use a lookup-table embedding scheme that preserves the temporal order of check-ins. Gao et al. (2017) introduced three TULER variants (TULER-G, TULER-L and BiTULER) combining this embedding approach with GRU (Cho et al., 2014), LSTM or bidirectional LSTM networks (Hochreiter & Schmidhuber, 1997). In subsequent works, various extensions of TULER were proposed, including TULVAE (Zhou et al., 2018) which combines an LSTM classifier with a variational autoencoder, and DeepTUL (Miao et al., 2020) which extends TULER with a historical attention module based on user IDs of previous check-ins sharing the same locations and time-slots.

With their advancement in other machine learning disciplines, newer studies on TUL have increasingly focused on transformer-based architectures. The T3S model (Yang et al., 2021) combines transformer and LSTM encoders to encode trajectories before classification. The purely transformer-based TULHOR (Alsaeed et al., 2023) embeds check-ins using hexagonal grids and supplements these with conventional POI embeddings.

Notably, several of these and other works have explored grid-based or multi-scale spatial encodings for trajectory and geographic tasks. T3S embeds locations by mapping them to grid cells, but uses only a single grid with fixed resolution, limiting its embeddings due to the trade-off between gradient density and specificity. TULHOR similarly relies on a single hexagonal grid level. Outside of TUL, DeeprETA (Hu et al., 2022) represents locations with geohashes at varying resolutions and combines them via feature hashing for time-of-arrival estimation, demonstrating the utility of capturing spatial information at multiple granularities. While it targets estimated time of arrival rather than TUL, we adapt its multi-resolution grid embedding to our online setting and include it as a baseline (Table 6). DeeprETA additionally applies feature hashing to bound the size of its location tables. This memory-bounding step is orthogonal and complementary to, rather than a replacement for, the multi-level sharing MiLES contributes (see Section 9). Hierarchical, multi-scale spatial grids are also an established approach in image geolocalization, where an image is classified into coarse-to-fine geographic cells (Weyand et al., 2016; Müller-Budack et al., 2018). A separate line of work encodes coordinates continuously rather than through discrete cells. Space2Vec (Mai et al., 2019) encodes geographic coordinates using multi-scale sinusoidal functions inspired by biological grid cells. Its *grid* variant represents each coordinate as interleaved sines and cosines whose wavelengths form a geometric series, while its *theory* variant applies the same multi-scale encoding along three directions offset by 60 degrees. This deterministic multi-scale encoding coincides with the positional-encoding form of Fourier features (Tancik et al., 2020), which has also been used in coordinate-based networks such as Neural Radiance Fields (Mildenhall et al., 2020). GeoCLIP (Cepeda et al., 2023) builds on the same idea but draws its frequencies from a random Gaussian matrix, scaling the frequency bandwidth across levels by an exponential schedule and summing the resulting per-level representations into a multi-resolution embedding. This exponential schedule governs the frequency bandwidth allocated to each level and combines them with equal weight, in contrast to MiLES's learnable per-level weighting and its geometric partitioning of the embedding dimensions across levels. These encodings share the same underlying principle of sinusoids spanning multiple spatial scales and differ mainly in whether the frequencies are fixed or sampled and in how the representation is trained. The original Space2Vec and GeoCLIP works pair their encoders with a pre-training stage, unsupervised and contrastively against images, respectively, which assumes access to the complete data distribution and is incompatible with our single-pass online setting. We therefore use the sinusoidal encoders directly, without a pre-trained network on top, to adapt them to the online learning scenario. We evaluate both the Fourier encoding, equivalently the Space2Vec grid variant, and the Space2Vec theory variant as baselines (Table 6), substituting them for MiLES's higher-level embeddings while retaining

the POI-based embedding at the base level. While these continuous approaches operate at the coordinate level, MiLES shares their principle of multi-scale spatial representation but applies it within the discrete lookup-table framework typical of TUL architectures. Specifically, MiLES contributes a learnable per-level scaling mechanism that aids optimization and a structured partitioning of the embedding dimensions in which coarser levels receive progressively fewer dimensions. This is detailed in Section 4 and is motivated by the notion that each embedding level sits at a different point on the sparsity-specificity trade-off and may therefore benefit from a different scale and capacity, a consideration not addressed by any of the aforementioned approaches.

Other notable TUL models have employed approaches include MainTUL (Chen et al., 2022), which combines transformers and RNNs through mutual distillation, and TGAN (Zhou et al., 2021c), which uses GANs for data augmentation. Further TUL models include GNNTUL (Zhou et al., 2021a) and AttnTUL (Chen et al., 2024), which process trajectories using graph neural networks, the self-supervised learning approach SML (Zhou et al., 2021b), as well as the Siamese neural network TULSN (Yu et al., 2020).

More broadly, trajectory and spatio-temporal modeling spans a range of modalities and tasks beyond check-in data, including vision-based pedestrian trajectory prediction (Zhong et al., 2023), spatio-temporal graph reasoning for social group detection (Dang et al., 2025), and multi-level deep feature fusion for state recognition (Ye et al., 2021). These works learn from continuous visual or sensor inputs, whereas we target discrete check-in sequences encoded through lookup-table embeddings over a large categorical POI vocabulary.

Although they address different aspects of the learning process and are therefore complementary to embedding techniques, general online machine learning techniques may also improve convergence of TUL models in an online setting, including replay methods (see e.g. Mnih et al., 2015; Lillicrap et al., 2019; Prabhu et al., 2020) and adaptive optimizers like Hypergradient Descent (Baydin et al., 2018) and DoG (Ivgi et al., 2023).

## 4  Multi-Level Spatial Embedding Sharing

Existing TUL approaches use lookup-table embeddings to encode the spatial information of check-ins. This embedding method uses a matrix $\boldsymbol{Z}$ of shape $|\mathbb{P}| \times d$, where $\mathbb{P}$ is the set of unique POIs and $d$ the embedding dimensionality. The embedding function can be denoted as $z(i) = \boldsymbol{Z}_i$, where $i$ is the index of a POI and $\boldsymbol{Z}_i$ is the $i$-th row vector of the embedding matrix. Given that only a single row is selected for any given location $\boldsymbol{l}$, the sparsity of this approach, defined as the fraction of active parameters in the embedding matrix, is simply $1 - 1/|\mathbb{P}|$. Since mobility data commonly includes thousands of unique POIs, the parameter usage and therefore also the gradients of such embeddings are generally very sparse. Existing methods, such as T3S (Yang et al., 2021) and TULHOR (Alsaeed et al., 2023), attempt to mitigate this issue by sharing embeddings between locations that belong to the same cell in a predefined spatial grid. By grouping multiple locations into shared embedding cells, the number of unique embeddings is reduced, thereby increasing parameter utilization and lowering sparsity to $1 - 1/|\mathbb{H}|$, where $|\mathbb{H}|$ is the number of cells in the grid-based partitioning of the coordinate plane.

This shared embedding approach creates a fundamental trade-off: while grouping locations into cells reduces gradient sparsity, it also reduces the informational content of the embeddings, as multiple distinct locations are now represented by a single vector. Embedding techniques that use a fixed level of sharing must therefore balance gradient density with representation specificity. We can formalize this information loss as a reduction in Shannon entropy (Shannon, 1948): when distinct locations are mapped to the same embedding, the model loses the ability to distinguish between them. Consider a subset $\mathbb{J}$ of location indices that correspond to at least two distinct locations within the same grid cell, and compare distinct against shared embeddings for these locations. When each location index $k \in \mathbb{J}$ has its own unique embedding $z(k)$, the contribution of these embeddings to the total information content is measured by their entropy:

$$H_{\text{distinct}} = -\sum_{k \in \mathbb{J}} p(z(k)) \log p(z(k)). \tag{3}$$

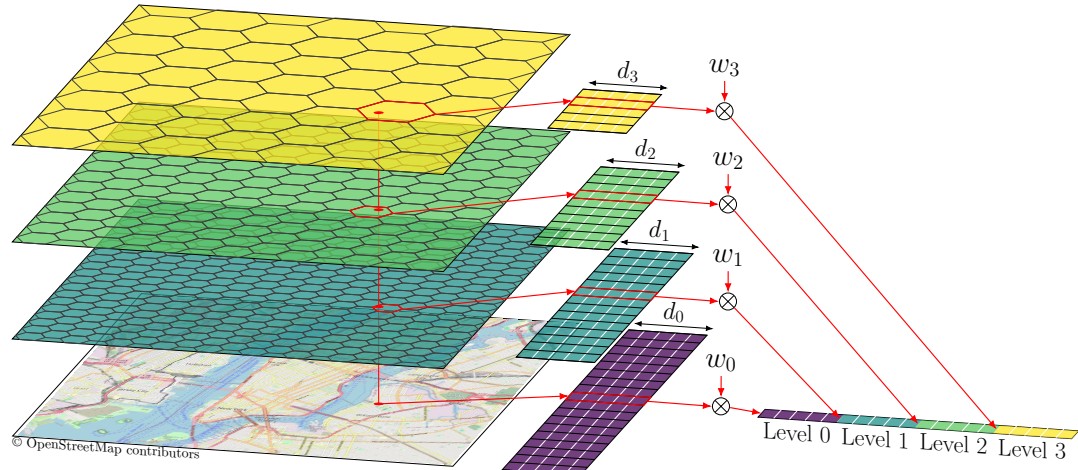

Figure 1: Visualization of the proposed multi-level spatial embedding sharing (MiLES) technique with learnable weights $w_l$ and embedding dimensions $d_l$ for each level $l \in \{0, 1, 2, 3\}$ and $\otimes$ representing scalar multiplication. Embedding dimensions are calculated according to Equation 6 but are drawn equally sized for visual clarity here.

When all location indices in $\mathbb{J}$ instead use a single shared embedding $\mathbf{z}^{(\text{share})}$, the probability mass concentrates on this single embedding, $p(\mathbf{z}^{(\text{share})}) = \sum_{k \in \mathbb{J}} p(z(k))$, and the entropy contribution becomes

$$H_{\text{shared}} = -\sum_{k \in \mathbb{J}} p(z(k)) \log \left( \sum_{k \in \mathbb{J}} p(z(k)) \right). \tag{4}$$

The information loss due to sharing is then given by the entropy difference

$$H_{\text{shared}} - H_{\text{distinct}} = -\sum_{k \in \mathbb{J}} p(z(k)) \log \left( \frac{\sum_{k \in \mathbb{J}} p(z(k))}{p(z(k))} \right). \tag{5}$$

Since $p(z(k)) > 0$ for all $k \in \mathbb{J}$ and $\sum_{k \in \mathbb{J}} p(z(k)) > p(z(k))$ for any individual term, the logarithm is strictly positive, so $H_{\text{shared}} < H_{\text{distinct}}$. Furthermore, as the number of locations in $\mathbb{J}$ increases, the sum $\sum_{k \in \mathbb{J}} p(z(k))$ grows larger, indicating greater information loss when more embeddings are aggregated within shared cells. Equivalently, writing $P = \sum_{k \in \mathbb{J}} p(z(k))$ for the total visitation mass of the cell and $q(k) = p(z(k))/P$ for the normalized within-cell distribution, the loss factorizes as $H_{\text{distinct}} - H_{\text{shared}} = P \, H(q)$. The loss is therefore largest when visits are spread uniformly across the locations in a cell, where $H(q)$ reaches its maximum $\log |\mathbb{J}|$, and shrinks as the distribution becomes more skewed. Since real-world POI visits are heavily long-tailed (Xu et al., 2024), most cells concentrate their visits on a few popular locations, so the loss stays well below this worst case, which helps explain why coarse sharing remains effective in practice. While this trade-off between gradient density and specificity is fixed when the data distribution is stationary, in an online setting the relative usefulness of density versus specificity may change as the stream progresses. For instance, after a concept drift occurs, such as the introduction of previously unobserved locations, rapid adaptation becomes crucial, while later on, more detailed, but slower-adapting, features may be preferable.

To address this challenge in online TUL, we propose multi-level spatial embedding sharing (MiLES), which generates embedding features that span a broad range of the density-information spectrum. MiLES is implemented as a drop-in replacement for standard embedding layers, adding minimal inference overhead while reducing the overall parameter count. In the following, we will give an in-depth description of the functionality of MiLES, which is also depicted in Figure 1. Like TULHOR's embedding approach (Alsaeed et al., 2023), MiLES maps locations to a grid created as a tiling of regular hexagons. We use hexagonal grids as they represent Euclidean distances more consistently than square grids (Ke et al., 2019). Unlike TULHOR, MiLES uses multiple mappings with increasing cell sizes and therefore increasing levels of aggregation.

Accordingly, we derive an index $h_l$ for each embedding-level $l \in \{0, 1, ..., l^{(\mathrm{max})}\}$. This index identifies the specific POI for $l = 0$ and the grid cell containing the check-in location for $l > 0$. For datasets without POI identifiers, such as GeoLife (Zheng et al., 2010), $l = 0$ corresponds to the finest-resolution grid instead.

To accommodate the emergence of previously unobserved locations, the embedding matrices $\boldsymbol{Z}_l$ can be dynamically expanded. To do so, a new randomly initialized row vector is appended to the matrix, whenever a check-in maps to an index $h_l$ that is not currently represented in $\boldsymbol{Z}_l$. This allows the model to adapt to new locations or grid cells as they appear in the stream without requiring a predefined vocabulary size. However, for ease of implementation in our experiments where the maximum grid indices are known a priori, we pre-allocate $\boldsymbol{Z}_l$ with shape $|\mathbb{H}_l| \times d_l$, where $|\mathbb{H}_l|$ is the number of unique location indices of level $l$. Since only embeddings indexed by the current input receive updates, pre-allocation yields behavior equivalent to dynamic expansion, provided the same parameter initialization is used. In a true deployment scenario where the spatial extent is not known in advance, dynamic expansion would be required, incurring a small per-insertion overhead for appending rows to the embedding matrices but avoiding the memory cost of pre-allocating for unseen regions.

To account for the decreasing informational content with increasing levels of aggregation, we assign smaller dimensions to higher-level embeddings. We compute the individual embedding dimensions as

$$d_l = \left\lfloor \frac{d \cdot \alpha^{-l}}{\sum_{l=0}^{l^{(\mathrm{max})}} \alpha^{-l}} \right\rfloor, \ \alpha > 1 \tag{6}$$

where $d$ is the dimension of the final embedding and $\alpha$ is a hyperparameter. To reach the total number of dimensions $d$ we add the remaining dimensions to the initial embedding level. Based on the results of a hyperparameter search (see Table 14), we use $\alpha = 2$ in our experiments.

For the final embedding, we concatenate all level-specific embeddings, each weighted by a learnable parameter $w_l$. Using $\|$ to represent vector concatenation, we define the embedding function $g$ as

$$g(\boldsymbol{h}; \boldsymbol{Z}_0, \boldsymbol{Z}_1, ..., \boldsymbol{Z}_{l^{(\mathrm{max})}}, \boldsymbol{w}) = \overset{l^{(\mathrm{max})}}{\underset{l=0}{\Big\|}} \boldsymbol{Z}_{l,h_l} \cdot w_l, \tag{7}$$

where $\boldsymbol{Z}_{l,h_l}$ is the row vector of the level-$l$ embedding table $\boldsymbol{Z}_l$ located at index $h_l$; the comma in the subscript separates the level index $l$ from the within-level row index $h_l$.

We select concatenation instead of summation to aggregate the level-specific embeddings to avoid interference between levels. For a fixed total embedding dimension, this approach also reduces the number of learnable parameters, since higher-level embeddings are shared across many locations.

Multiplying each level-specific embedding $\boldsymbol{Z}_{l,h_l}$ by a learnable scalar $w_l$ does not increase the capacity of the model. Because the embedding tables $\boldsymbol{Z}_l$ and the backbone's input weights are learnable, they can absorb any rescaling of $w_l$: for any nonzero scalar $c$, replacing $w_l$ by $c \, w_l$ and $\boldsymbol{Z}_l$ by $\boldsymbol{Z}_l/c$ leaves the forward map unchanged. A single $w_l$ therefore adds no representational capacity and cannot be read on its own as the importance of a spatial level. As we use no weight decay, this symmetry is exact. Instead, the weights play an optimization role. A single scalar per level lets the model rescale the contribution of an entire embedding table through one parameter, which we find aids online learning when each table is updated only sparsely. The weights are updated by gradient descent alongside all other parameters and, as we show in Figure 6, tend to grow over training.

To embed a full trajectory $T = (\boldsymbol{c}_1, \ldots, \boldsymbol{c}_n)$, this process is applied independently to each check-in: the location $\boldsymbol{l}_i$ is passed through MiLES to obtain a spatial embedding $\boldsymbol{g}_i$, while the timestamp $t_i$ is encoded via an hour-of-day lookup embedding. These are concatenated to form the check-in representation, which serves as a drop-in replacement for the spatial input of any sequence-based TUL backbone. The sequence of check-in representations is then processed by the backbone (e.g., BiTULER's bidirectional LSTM) to produce a trajectory-level representation for user classification. All parameters including MiLES's embedding matrices $\boldsymbol{Z}_l$, level weights $\boldsymbol{w}$, temporal embeddings, and backbone parameters are trained jointly via the loss in Equation (2). Algorithm 1 in Appendix A.1 summarizes MiLES's per-check-in embedding construction.

# 5 Experiments

To evaluate the impact of the proposed MiLES approach and its individual components on the performance of existing TUL models in a data stream setting, we perform a series of experiments.

We use the widely adopted Foursquare-NYC, Foursquare-TKY (Yang et al., 2015) and GeoLife (Zheng et al., 2010) datasets. Following standard preprocessing (Chen et al., 2022), we split each trajectory into shorter segments with a maximum length of 24 hours for Foursquare-NYC and Foursquare-TKY and 3 hours for GeoLife, and selecting the most active users. For GeoLife, we additionally subsample check-ins at one-minute intervals. For more information on the selected datasets, see Table 1.

For optimization, we use Adam (Kingma & Ba, 2017). Across datasets, the input modalities include GPS coordinates, timestamps, and POI identifiers, except in GeoLife, which lacks POI information. We also provide hour-specific lookup embeddings, following prior work (see, e.g. Chen et al., 2022; Miao et al., 2020). We do not incorporate components that rely on data beyond the trajectories themselves (e.g., TULHOR's mobility flows), as such information is absent from standard trajectory datasets and lies outside the scope of our embedding-based approach.

We tune all hyperparameters by maximizing mean prequential top-1 accuracy on the first 5,000 trajectories from the 400-user Foursquare-TKY stream. This procedure is applied consistently for all models and methods including MiLES, with tuning data excluded from evaluation. The selected configuration is then frozen for the remainder of the stream, so tuning relies only on the earliest trajectories and never on future or held-out data. We deliberately restrict tuning to a small initial segment from a single dataset to reflect the constraints of a realistic online deployment, where labeled data for hyperparameter optimization from the same data source is often times unavailable at the outset. Using these same hyperparameters for all datasets without adjusting them means that no method is optimized individually for each experimental condition, and any disadvantage from cross-dataset transfer affects all configurations equally. The comparison of online and batch learning scenarios in Section 6.1 uses a separately tuned hyperparameter configuration, under which MiLES's gains remain comparable in magnitude (see Table 10) on all datasets including GeoLife. This suggests that the results are not sensitive to the specific hyperparameter setting and that the shared protocol does not disproportionately affect any method.

For methods requiring historical data (MainTUL and DeepTUL), we maintain a buffer of the last 1,000 trajectories. To determine embedding sharing levels, we partition the map into multiple hexagonal grids and tune the number of rows using the same protocol. The best configuration uses three levels with 200 rows at the base, halving the number of rows at each subsequent level. For GeoLife, POI embeddings are replaced with a grid-based embedding of 800 rows.

A complete list of hyperparameters is provided in Table 14. To ensure fairness, the total embedding dimensionality is fixed across methods (after baseline tuning), so that performance differences reflect only the embedding strategy rather than representational capacity.

To address the impact of MiLES's hyperparameters, we conduct a detailed analysis of the effect of the number of embedding levels and grid resolutions on model performance, as shown in Figure 5. Standard deviations are omitted from tables for brevity. Unless stated otherwise, the reported results represent averages over 5 independent runs.

Table 1: Datasets used for experimental evaluation.

| Dataset | Foursquare-NYC | | Foursquare-TKY | | GeoLife | |
|---|---|---|---|---|---|---|
| Users | 800 | 400 | 800 | 400 | 150 | 75 |
| Trajectories | 61,218 | 35,510 | 70,007 | 44,955 | 25,611 | 23,290 |
| Check-Ins | 196,435 | 137,886 | 324,564 | 248,771 | 1,284,208 | 1,187,510 |
| POIs | 34,383 | 25,443 | 38,212 | 28,286 | — | — |

# 6 Results

In this section, we present a comprehensive evaluation of MiLES to validate its effectiveness and robustness in online learning scenarios. Our analysis is structured as follows: First, we demonstrate the practical benefits of MiLES by integrating it into several state-of-the-art TUL models. Following this, we compare its performance against alternative embedding strategies and general online learning techniques. Finally, we conduct an in-depth analysis of the MiLES architecture itself, using a detailed ablation study and statistical tests to isolate and verify the contribution of each of its individual components.

**Enhancing TUL Models with MiLES** We evaluated various TUL approaches with their original location embedding techniques for online learning applications. The results of these experiments for the higher dataset variants with higher user counts are depicted in the upper section of Table 2 (see Table 17 for all results). As Table 2 shows, the bidirectional-LSTM-based BiTULER achieved the overall best performance on all datasets, except for GeoLife where DeepTUL yielded a higher top-1 accuracy and macro F1 score. This exception likely stems from BiTULER being the most lightweight model with the fewest parameters, enabling faster adaptation. Similarly, the generally lower performance of the transformer-based models (MainTUL, T3S, and TULHOR) compared to their RNN-based counterparts may reflect the challenges of adapting more complex architectures in online learning scenarios. We then replaced the original embedding modules in each TUL model with MiLES, keeping the total embedding dimension constant. The bottom section of Table 2 shows the relative change in performance metrics measured in percentage points (pp.) achieved by using MiLES.

The largest improvements were on GeoLife, where MiLES increased top-1 accuracy by up to 8.72 pp. (Bi-TULER) and top-5 accuracy by up to 6.98 pp. On the POI-rich Foursquare datasets the gains are smaller but consistent, with relative top-1 improvements of roughly 2 to 5%. This larger gain for GeoLife likely stems from its lack of POI information, where MiLES's additional embedding levels provide particular value compared to existing techniques' original embedding approach. The improvements themselves are highly stable across paired random seeds (standard deviation at most 0.78 pp. for Δ MiLES; see Table 18), and MiLES improves over the default embedding in every one of the 18 model × dataset settings on every metric. Even the noisiest of these, MainTUL's macro F1 gain on GeoLife, averages 5.50 pp., about seven times its 0.78 pp. standard deviation. Per-model paired significance tests are reported in Table 19, where a one-sided Wilcoxon signed-rank test confirms a significant improvement for each model, dataset and metric individually. Notably, improvements were consistent across all models, including T3S and TULHOR, which already incorporated single-level grid-based embeddings, demonstrating the consistent benefits of MiLES's multi-level approach in online learning settings.

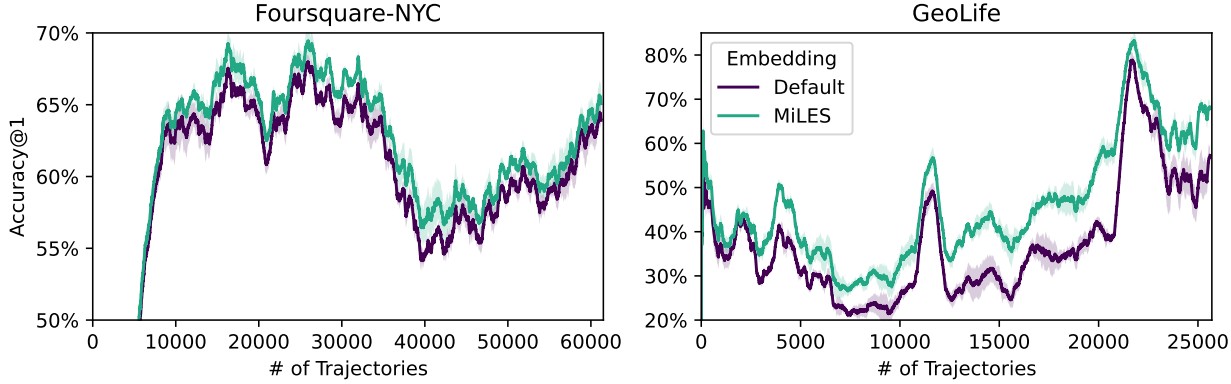

Figure 2: Rolling top-1 accuracy for BiTULER with and without MiLES on Foursquare-NYC (window size 5,000) and GeoLife (window size 1,000). Shaded areas indicate the $4\sigma$ interval over all random seeds. The increase in accuracy toward the end of the GeoLife stream coincides with a reduction in diversity of user labels within the evaluation window, which lowers the effective difficulty of the classification task in this segment (see Figure 14).

Table 2: Top-1 accuracy, top-5 accuracy and macro F1 score [%] averaged over prequential evaluation runs for TUL models with their original embedding technique (Default) and performance gains when using our technique instead (Δ MiLES).

| Dataset | Foursquare-NYC | | | Foursquare-TKY | | | GeoLife | | |
|---|---|---|---|---|---|---|---|---|---|
| Model | Acc@1 | Acc@5 | F1 | Acc@1 | Acc@5 | F1 | Acc@1 | Acc@5 | F1 |
| Default | | | | | | | | | |
| BiTULER | **60.12** | **67.20** | **57.83** | **63.16** | **74.91** | **61.06** | **37.56** | 70.85 | 26.69 |
| TULVAE | 59.79 | 66.77 | 57.32 | 55.82 | 66.23 | 51.68 | 37.08 | 70.45 | 25.25 |
| DeepTUL | 58.72 | 65.48 | 56.60 | 61.17 | 72.49 | 59.10 | 36.32 | **72.64** | **29.82** |
| MainTUL | 55.67 | 62.61 | 53.01 | 59.53 | 71.89 | 57.09 | 34.00 | 70.26 | 21.76 |
| T3S | 52.98 | 60.28 | 49.50 | 56.41 | 69.26 | 53.24 | 35.25 | 71.11 | 21.52 |
| TULHOR | 53.85 | 61.13 | 50.40 | 56.61 | 69.61 | 53.45 | 34.65 | 72.46 | 24.92 |
| Δ MiLES | | | | | | | | | |
| BiTULER | +1.49 | +3.58 | +1.71 | +1.40 | +2.85 | +1.30 | +8.72 | +6.98 | +6.26 |
| TULVAE | +1.79 | +3.73 | +2.04 | +3.07 | +4.43 | +3.34 | +8.01 | +6.57 | +4.74 |
| DeepTUL | +1.06 | +3.04 | +1.26 | +0.84 | +2.37 | +0.77 | +8.59 | +6.31 | +5.97 |
| MainTUL | +1.44 | +4.33 | +1.52 | +1.62 | +3.45 | +1.45 | +8.19 | +6.18 | +5.50 |
| T3S | +1.71 | +3.13 | +2.10 | +1.70 | +2.79 | +1.91 | +6.98 | +4.80 | +5.18 |
| TULHOR | +1.71 | +3.16 | +2.11 | +1.85 | +2.80 | +2.03 | +8.20 | +5.03 | +5.68 |

Furthermore, the embedding dimension was fixed at a single value tuned for the default embedding, so the comparison does not favor MiLES. At this dimension MiLES in fact uses fewer embedding parameters than the default (Table 16), so its gains are not obtained through additional capacity.

In addition to these aggregate results, Figure 2 shows the evolution of rolling top-1 accuracy over the data stream for BiTULER with and without MiLES on Foursquare-NYC and GeoLife. On Foursquare-NYC, MiLES maintains a relatively consistent advantage throughout the stream rather than concentrating its gains in any particular segment, suggesting that the benefits of multi-level sharing persist as the model accumulates training signal. The gap appears to widen slightly during the accuracy dip around trajectory 40,000, which coincides with elevated distributional shift in POI visits (see Figure 13), suggesting that MiLES may be particularly beneficial during periods of non-stationarity. On GeoLife, the advantage of MiLES grows over the course of the stream, consistent with the shared grid-level embeddings accumulating increasingly informative spatial representations as more locations are observed.

To evaluate the impact of MiLES on model behavior after a concept drift, we removed all check-ins in the eastern half of the Foursquare-NYC dataset from the first 30,000 trajectories, so that the model would suddenly encounter a large fraction of previously unobserved POIs. This emulates an abrupt concept drift that may occur, for instance, if a location-based service expands to cover a larger area. Table 3 quantifies the novelty introduced by this intervention at each MiLES embedding level. At the POI level, over a third of check-ins in the first 1,000 post-drift trajectories reference entirely new locations, while at the coarsest

Table 3: Novelty introduced by the simulated concept drift on Foursquare-NYC at each MiLES embedding level. *New locations* are indices (POIs or grid cells) encountered after the drift that were absent from the pre-drift stream. The rightmost column restricts this count to the first 1,000 post-drift trajectories.

| Level | Total Locations | New Locations (Total) | New Locations (First 1k Tr.) |
|---|---|---|---|
| 0 | 28,214 | 12,321 | 745 |
| 1 | 7,549 | 3,743 | 360 |
| 2 | 4,129 | 2,012 | 269 |
| 3 | 1,716 | 804 | 177 |

Table 4: Pre- and post-drift performance for BiTULER with and without MiLES on the simulated Foursquare-NYC drift experiment. Rolling accuracy is computed with a window of 1,000 trajectories. Pre-drift performance is the rolling accuracy immediately before the drift. Recovery denotes the number of post-drift trajectories until rolling accuracy returns to 100% of the pre-drift value. Avg. during recovery refers to the mean rolling accuracy over this post-drift-to-recovery window.

| | Pre-drift [%] | | Post-drift min. [%] | | Recovery [trajectories] | | Avg. during recovery [%] | |
|---|---|---|---|---|---|---|---|---|
| | Acc@1 | Acc@5 | Acc@1 | Acc@5 | Acc@1 | Acc@5 | Acc@1 | Acc@5 |
| Default | 52.94 | 60.76 | 36.68 | 42.68 | 2,032 | **2,112** | 45.42 | 52.43 |
| MiLES | **55.68** | **65.30** | **39.44** | **46.90** | **1,760** | 2,197 | **47.01** | **56.93** |

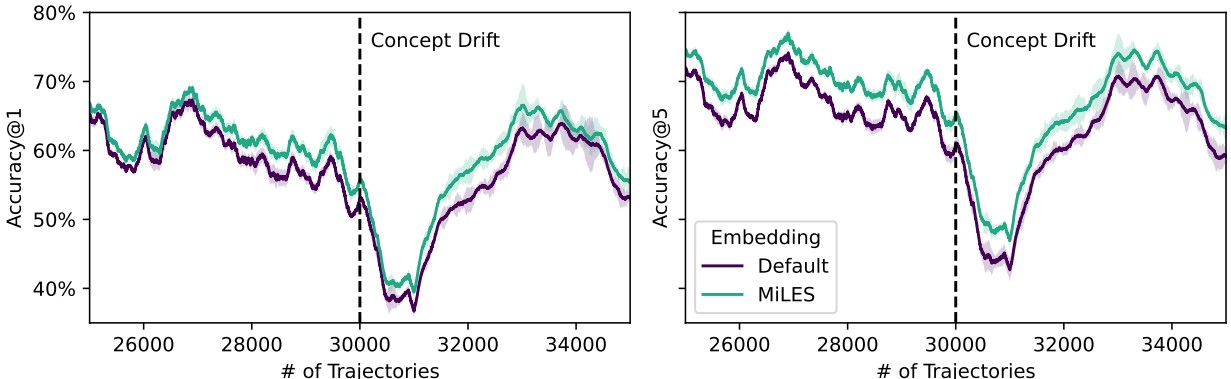

Figure 3: Impact of simulated spatial expansion on Foursquare-NYC. After 30000 trajectories, the model was suddenly exposed to check-ins in the eastern half of the city, which were excluded from the initial trajectories. Both metrics were calculated with a sliding window of size 1000. The highlighted areas mark the $2\sigma$ interval over all random seeds.

grid level this fraction drops below a quarter. This illustrates how MiLES could mitigate the impact of novel locations. While POI-level embeddings must be learned from scratch, coarser levels already carry informative spatial representations from nearby, previously observed check-ins.

The performance impact is shown in Figure 3 and quantified in Table 4. MiLES starts from a higher pre-drift accuracy and maintains a higher performance floor after the drift across both metrics. To reach its pre-drift top-1 accuracy, MiLES requires roughly 13% fewer trajectories, whereas its top-5 accuracy recovers about 4% slower, a small difference relative to the 1,000-trajectory rolling window used to compute it. The average accuracy during the recovery window is nonetheless consistently higher for MiLES on both metrics, indicating that its advantage holds throughout the recovery period rather than only at the post-drift minimum. This is consistent with the novelty analysis: the coarser grid levels provide a more informed starting point for newly introduced locations, raising the minimum but not fundamentally accelerating learning at the POI level.

Finally, we analyzed the computational efficiency of MiLES using the BiTULER model. As detailed in Table 5, MiLES reduces the model size, decreasing the number of embedding parameters by approximately 38% on the Foursquare datasets (e.g., from 35.2M to 22.0M on NYC). Table 16 gives the full per-level breakdown. Furthermore, the lower parameter counts translate to faster model updates and therefore shorter runtimes per online learning iteration, due to the more efficient updates outweighing MiLES's small inference overhead. Since the parameter count of the default embeddings on Geolife is already relatively low, we observe a slight increase in wall time for this dataset. GeoLife trajectories are also considerably longer on average than those in the Foursquare datasets (Table 1), so MiLES's per-check-in overhead of multiple lookups and aggregation is incurred more often per trajectory, further contributing to this effect. This increase could potentially be mitigated with an optimized implementation of MiLES.

Table 5: Number of embedding parameters and wall time per iteration for BiTULER using MiLES or its original embedding technique. Measured on a system with an Intel i5-9600K CPU and an Nvidia RTX 3090 GPU.

| Dataset | Foursquare-NYC | | Foursquare-TKY | | GeoLife | |
|---|---|---|---|---|---|---|
| Embedding | # Parameters | Wall Time | # Parameters | Wall Time | # Parameters | Wall Time |
| Default | 35.21M | 7.99 ms | 39.13M | 8.50 ms | 4.74M | **11.27 ms** |
| MiLES | **21.97M** | **7.38 ms** | **24.32M** | **7.81 ms** | **3.28M** | 11.66 ms |

**Comparison With Alternative Embeddings and Online Learning Techniques**  We further evaluated our approach against alternative embedding-, experience replay- and adaptive optimization methods. The results, averaged across all models from Table 2, are shown in Table 6. Among the embedding techniques, MiLES achieved the best overall performance on the POI-rich Foursquare datasets across all metrics. It consistently outperformed the baseline POI-based lookup embeddings, as well as the hybrid linear and Fourier embeddings. These two alternatives combined POI-based embeddings with either a learnable linear projection or a Fourier feature encoding (Tancik et al., 2020) of the location coordinates, replacing the higher-level embeddings used in MiLES. Following standard practice, the Fourier encoding maps each coordinate through sines and cosines within a fixed range of geometrically spaced frequencies, which corresponds to the *grid* variant of the Space2Vec encoder (Mai et al., 2019). The POI lookup and the coordinate encoding each receive half of the embedding dimensions, and the frequency scale is the only tunable parameter, tuned with the same prequential procedure used for every other method (Table 14). On the Foursquare datasets, the linear projection underperformed the POI-only embedding and the Fourier encoding roughly matched it, but neither reached MiLES, indicating that on POI-rich data discrete multi-level sharing uses the embedding budget more effectively than a continuous coordinate encoding.

Table 6: Top-1 accuracy, top-5 accuracy and macro F1 score [%] of different embedding-, experience replay- and adaptive optimization methods, averaged across all models in Table 2 (full results in Table 21). The default uses POI-based lookup embeddings and Adam without replay; each method replaces one default component. Linear, Fourier and Space2Vec embeddings concatenate a POI lookup with a coordinate encoding (linear projection, or sinusoidal basis, respectively), each contributing half the embedding dimensions; Fourier and Space2Vec are the *grid* and *theory* variants of the Space2Vec encoder (Mai et al., 2019). DeeprETA denotes a DeeprETA-style embedding (Hu et al., 2022) with a multi-resolution square grid, uniform per-level dimensions, and no learnable level weights.

| Dataset | Foursquare-NYC | | | Foursquare-TKY | | | GeoLife | | |
|---|---|---|---|---|---|---|---|---|---|
| Method | Acc@1 | Acc@5 | F1 | Acc@1 | Acc@5 | F1 | Acc@1 | Acc@5 | F1 |
| Default | 56.85 | 63.91 | 54.11 | 58.78 | 70.73 | 55.94 | 35.81 | 71.30 | 25.00 |
| *Embedding* | | | | | | | | | |
| Linear | 50.74 | 59.21 | 47.51 | 53.48 | 66.25 | 50.16 | 35.02 | 70.62 | 22.66 |
| Fourier | 56.62 | 66.80 | 53.99 | 57.09 | 71.44 | 53.92 | **54.58** | **84.33** | **36.70** |
| Space2Vec | 55.18 | 65.98 | 52.47 | 58.01 | 73.10 | 54.97 | 53.26 | 83.99 | 35.64 |
| DeeprETA | 56.55 | 66.58 | 53.84 | 56.84 | 71.06 | 53.77 | 44.43 | 78.49 | 30.63 |
| MiLES | **58.39** | **67.41** | **55.90** | **60.53** | **73.85** | **57.74** | 43.93 | 77.27 | 30.55 |
| *Replay* | | | | | | | | | |
| FIFO | 57.38 | 64.53 | 55.24 | 59.45 | 71.26 | 56.98 | 37.33 | 70.87 | 25.82 |
| Balanced | 57.19 | 64.26 | 54.72 | 58.75 | 70.48 | 55.80 | 36.40 | 68.37 | 25.36 |
| FIFO+MiLES | **58.94** | **67.88** | **56.87** | **61.17** | **74.19** | **58.68** | **45.86** | **77.49** | **31.87** |
| *Optimizer* | | | | | | | | | |
| DoG | 36.79 | 42.57 | 33.82 | 35.95 | 45.74 | 34.00 | 29.12 | 60.59 | 21.14 |
| AdamHD | 45.94 | 51.80 | 43.18 | 40.15 | 49.03 | 37.89 | 22.24 | 48.18 | 17.09 |

On the POI-free GeoLife dataset, however, the Fourier encoding was the strongest embedding, outperforming MiLES, as continuous encoders appear better suited to GeoLife's denser, continuous GPS data (Table 8). We additionally evaluated the *theory* variant of Space2Vec, which uses a multi-directional rather than grid-aligned sinusoidal basis and performed within roughly 1.5 pp. of the Fourier encoding on every dataset. We also compared against a DeeprETA-style embedding (Hu et al., 2022), which replaces MiLES's variable per-level dimensions and learnable level weights with a uniform-dimension, unweighted concatenation over a multi-resolution square grid. On the Foursquare datasets, MiLES outperformed it across all backbones. On GeoLife, which lacks POI features and thus relies entirely on grid-based embeddings, the DeeprETA-style embedding was competitive with MiLES and slightly ahead in the aggregate, though both trailed the continuous Fourier encoder. This reflects our deliberately realistic tuning protocol. Because no in-distribution data is available before the stream, we tune each method once and transfer its structural grid configuration to all datasets. Under this single-transfer regime, DeeprETA's tuned grid happens to align with GeoLife's larger spatial extent more favorably than MiLES's does. A controlled comparison that holds the grid fixed (Appendix B.3) confirms that MiLES's embedding design remains advantageous on GeoLife as well, showing that this gap reflects grid transfer rather than the embedding design itself. For memory replay strategies, we compared a FIFO buffer and a random class-balanced buffer, each limited to 1,000 past trajectories, with one sample replayed per training step. Both replay strategies improved performance over the default configuration with the models original embeddings and without replay. Notably, the FIFO buffer consistently outperformed the class-balanced one, suggesting that data recency is more important than class balance in the non-stationary streaming setting. When combining MiLES with FIFO replay (FIFO + MiLES), we observed further gains across all datasets and metrics, demonstrating that MiLES remains effective when paired with replay techniques. The adaptive optimizers DoG and AdamHD performed worse than the standard Adam optimizer, used for all other methods. We attribute this to the high gradient variance introduced by training on individual samples instead of mini-batches, which destabilizes the learning rate adaptation in both methods.

**Analysis of MiLES Components**    To assess the contribution of individual components in MiLES, we conducted ablation studies by systematically removing embedding levels (-L1, -L2, -L3), the learnable weighting parameters $w_l$ (-WL), and the level-dependent embedding dimensions (-VD). When excluding an embedding level, we maintained the total embedding dimension and adjusted the dimensions of the remaining levels by omitting the affected level from Equation 6. In the -VD setting, we distributed the total embedding size equally among levels.

Figure 4 plots the macro F1 scores and top-5 accuracy of our full approach and each ablation, revealing that the complete embedding is consistently on or near the Pareto frontier and achieves a favorable balance between the two metrics. This is particularly evident in the average performance across all datasets, where removing any component results in a noticeable drop in at least one metric. For instance, removing the variable dimensionality (-VD)

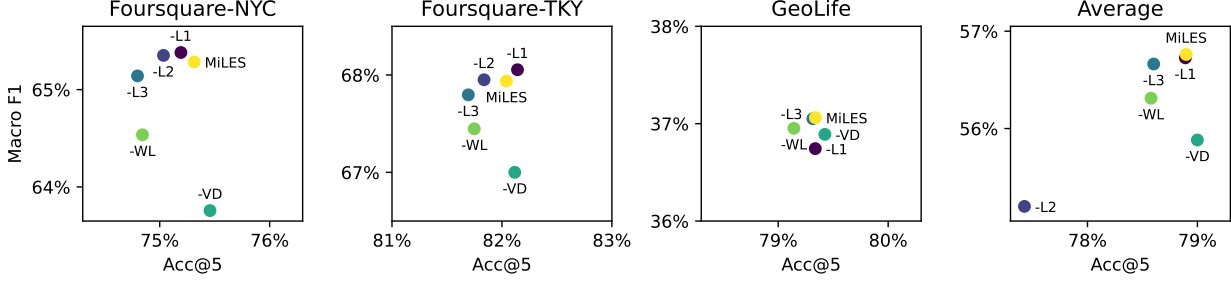

Figure 4: Macro F1 score and top-5 accuracy results of ablation study (-L1/-L2/-L3 remove the corresponding embedding level, -VD removes the level-dependent embedding dimensions, -WL removes the learnable level weights $w_l$) for MiLES with BiTULER as the underlying model. Metrics are averaged over both variants of each dataset. -L2 was omitted in the GeoLife and average results for visual reasons, since the removal of the level two embeddings resulted in a large decrease in top-5 accuracy on GeoLife.

Table 7: P-values of one-sided Wilcoxon signed-rank tests, with the null hypothesis that removing the respective component does not degrade the respective performance metric. Each paired observation is the metric difference (full MiLES minus ablated) from a single prequential evaluation run, identified by a unique combination of dataset variant and seed. With 2 variants and 10 seeds per dataset, each test is based on N = 20 paired observations. Significant values are highlighted, with higher significance levels receiving a darker hue.

| Dataset | Foursquare-NYC | | | Foursquare-TKY | | | GeoLife | | |
|---|---|---|---|---|---|---|---|---|---|
| Method | Acc@1 | Acc@5 | F1 | Acc@1 | Acc@5 | F1 | Acc@1 | Acc@5 | F1 |
| -L1 | 0.996 | <0.001 | 0.992 | 0.998 | 0.999 | 0.996 | 0.022 | 0.712 | 0.003 |
| -L2 | 0.938 | <0.001 | 0.973 | 0.411 | <0.001 | 0.727 | <0.001 | <0.001 | <0.001 |
| -L3 | <0.001 | <0.001 | <0.001 | <0.001 | <0.001 | <0.001 | <0.001 | 0.273 | 0.364 |
| -VD | <0.001 | 0.969 | <0.001 | <0.001 | 0.999 | <0.001 | 0.152 | 0.905 | 0.101 |
| -WL | <0.001 | <0.001 | <0.001 | <0.001 | <0.001 | <0.001 | <0.001 | 0.001 | 0.147 |

leads to a large decrease in macro F1 for only a marginal gain in top-5 accuracy. Only the removal of the highest resolution grid embeddings (-L1) results in an insignificant drop in average performance metrics. The dataset-specific plots show that the highest-resolution grid (-L0) offers no additional benefit on Foursquare-TKY, likely due to the datasets high density of check-ins per POI. In all other scenarios, however, each component proves its value.

To validate these observations statistically, we performed one-sided Wilcoxon signed-rank tests (Table 7). For each ablation, dataset and metric, we constructed paired observations by matching the full MiLES configuration against the ablated configuration for the same combination of dataset variant and seed. This yielded 20 pairs per test (two dataset variants × 10 random seeds). The one-sided test evaluates whether the paired differences (full MiLES minus ablated) are systematically positive against the null hypothesis that these differences are distributed symmetrically around zero.

On the POI-rich Foursquare datasets, the additional embedding levels primarily yield highly significant gains in top-5 accuracy ($p < 0.001$). Conversely, on GeoLife, which lacks POI data, these levels provide statistically significant improvements to top-1 accuracy and macro F1 score. Similarly, the variable dimensionality (-VD) and learnable weights (-WL) offer highly significant contributions to performance across most datasets and metrics.

In conclusion, while the performance gain from any single component may seem moderate in isolation, our analysis shows that all components contribute significantly to the model's overall robustness. It is also worth noting that these gains are conservative in a capacity sense. Because the total embedding

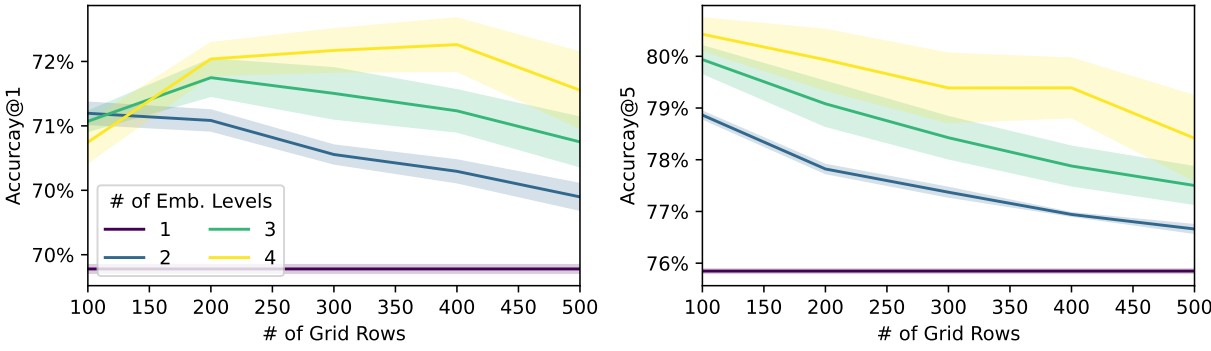

Figure 5: Performance on the 400 user Foursquare-NYC dataset, relative to the number of embedding levels (color) and rows in the base-level grid (x-axis) using BiTULER and MiLES with fixed level weights $w_l$. Shaded areas represent the $1\sigma$ range.

dimension was kept fixed for a fair comparison, adding higher-level embeddings reduces the dimension available to the POI-level embedding, and Table 8 shows that reducing POI-level capacity alone degrades performance. Each embedding levels gain therefore has to overcome this capacity loss to register as a net improvement. Furthermore, even a component like L1, whose average gain is small, is justified by its negligible computational cost of a single table lookup, as it provides robustness on datasets like GeoLife without sacrificing efficiency.

In another experiment, we evaluated MiLES for varying embedding aggregation levels and varying grid resolutions with the level-specific weights $w_l$ fixed at 1. For this, we used the Foursquare-NYC dataset with 400 users and BiTULER as the underlying classifier. Following previous experiments, level two used grids at half the base resolution and level three at quarter resolution. The results in Figure 5 indicate that increasing the number of embedding levels improves performance, provided that the base-level grid has a sufficiently high resolution. In particular, the four-level embedding module achieves the highest top-1 and top-5 accuracy when the base-level grid contains at least 200 rows. This suggests that MiLES is robust to variations in base grid resolution and that all embedding levels contribute meaningfully to performance. The effect of base grid resolution differs between top-1 and top-5 accuracy. While top-1 accuracy peaks at 400 grid rows for the four-level configuration, top-5 accuracy decreases steadily as resolution increases. This is consistent with earlier observations that coarser sharing benefits group-level prediction and therefore top-5 accuracy.

To examine the optimization dynamics induced by the level weights $w_l$, we tracked their values across multiple prequential evaluation runs on the 400-user Foursquare-NYC dataset. Figure 6 shows the evolution of these weights alongside the number of unique users in the 1,000 most recent trajectories.

On average, the weights of all levels increase throughout training, which rescales the level contributions. This is consistent with the optimization role described in Section 4. The weights also co-move with shifts in user diversity, contracting as the number of unique users in the recent window decreases after about 15,000 trajectories. However, their relative magnitudes cannot be interpreted as direct measures of level importance because each weight and its embedding table can be rescaled against one another without altering the model (Section 4). As we show next, once the embedding magnitudes and the backbone's read-out are folded in, the POI level in fact dominates the per-level contribution throughout the stream, even though $w_3$ is the largest weight. The weights are therefore best understood as a per-level optimization control rather than a measure of which spatial granularity matters most. Because the weights are non-identifiable, the consistent improvement they provide (Table 7) cannot reflect added capacity, and must instead come from how the parametrization shapes the online optimization. Figure 7 makes this concrete. For each level we track its forward contribution magnitude $g_l = w_l \|\boldsymbol{Z}_l\| \|\boldsymbol{W}_l\|$, the product of the level weight $w_l$, the embedding norm $\|\boldsymbol{Z}_l\|$, and the norm $\|\boldsymbol{W}_l\|$ of the backbone input weights that read that level. Because a level's overall scale can be moved freely among these three factors without changing the model, $g_l$ does not depend on how the scale is split, and is comparable both across levels and between models. Both models begin from the same

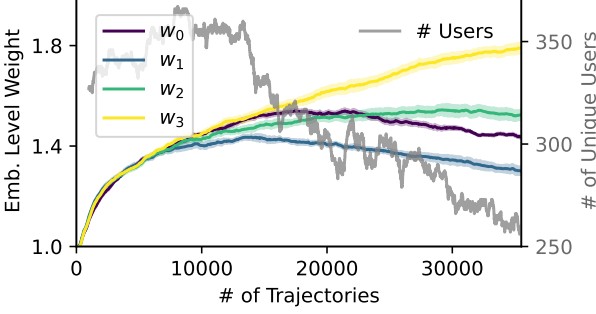

Figure 6: Mean learnable level weights $w_0, \ldots, w_3$ (left axis) and the number of unique users within the last 1,000 trajectories (right axis) on 400-user Foursquare-NYC. All weights grow over training, the coarsest ($w_3$) largest, and co-move with user diversity. Shaded areas represent the $1\sigma$ range over five seeds.

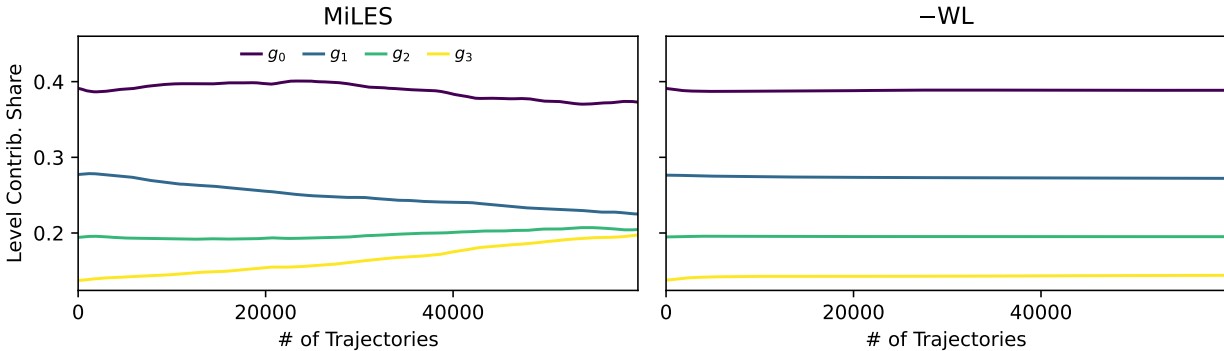

Figure 7: Per-level forward-contribution share $g_l / \sum_l g_l$ over the online stream on Foursquare-NYC (800 users, BiTULER, mean over five seeds), where $g_l = w_l \, \|\boldsymbol{Z}_l\| \, \|\boldsymbol{W}_l\|$ is the total magnitude at which level $l$ enters the backbone.

dimension-dependent profile in which the POI level dominates. Over training, MiLES equalizes the shares of the three grid-based levels, raising the coarsest level, while the POI share slowly declines after about 20,000 trajectories. This shift comes almost entirely from $w_l$, while the embedding and input-weight norms barely differ across levels throughout the stream (see Figure 11). Without the level weights, the same re-weighting would require coordinated changes across the many input weights of each level, which $-$WL does not achieve. Its profile stays close to its initial $g_l$, even though the gradient appears to push the coarsest input block to grow slightly. Overall, the level weights likely help in terms of model performance because they give the optimizer a single parameter controlling each level's forward scale, whereas reaching the same scale through the high-dimensional input block is far slower. A complementary check rules out the alternative reading that $w_l$ acts as an effective per-level learning rate rather than a forward scale (Appendix B.6).

**Embedding Capacity and Cell-Diversity Diagnostics**   Two questions concern the embedding capacity. First, whether MiLES's gains are partly explained by a reduction in POI-level capacity. Since MiLES allocates a portion of the total dimension to coarser levels (Equation (6)), the POI level receives fewer dimensions than in the single-level baseline. Second, how the methods compare as the *total* embedding dimension, the overall budget we otherwise hold fixed for a fair comparison, varies. We address both with a single sweep of the total location embedding dimension from 128 to 2048, holding the recurrent hidden size fixed and reusing the remaining hyperparameters tuned at $d = 1024$ for all methods, and compare MiLES against the single-level POI lookup and a Fourier-feature encoder (Table 8).

On the capacity question, reducing the POI lookup's dimension monotonically *degrades* performance (top-1 falls from 60.11% at $d = 1024$ to 44.93% at $d = 128$), so implicit regularization from reduced POI capacity cannot explain MiLES's gains. MiLES allocates only 547 of its 1024 dimensions to level 0 (Table 16), yet reaches 61.60% top-1 accuracy, surpassing the single-level baseline that assigns all 1024 dimensions to POI embeddings. The coarser levels therefore more than compensate for the reduced POI-level capacity, confirming that the gains arise from the additional spatial information they provide.

On the budget question, MiLES leads at every dimension and on every metric. Its advantage over the POI lookup is largest when the total embedding dimension is small, reaching 8.6 pp. at $d = 128$, where spreading a small dimension budget across shared coarse levels appears to use the limited capacity more effectively than assigning it all to individually-learned POI embeddings. Viewed on parameters rather than dimension, MiLES at $d = 1024$ (22.0M embedding parameters, Table 16) already matches the POI lookup at $d = 2048$ (61.60% vs. 61.69% top-1) while using under one-third of the baseline's 70.4M embedding parameters, so its gains are not bought with additional capacity. The ranking holds across the full range rather than being an artifact of the chosen dimension. Although MiLES attains its highest absolute accuracy at $d = 2048$, we report our main experiments at $d = 1024$, the largest dimension in our tuning grid. Doubling the dimension nearly doubles the embedding parameters and slows each update for diminishing returns. MiLES top-1 rises 1.49 pp. from $d = 512$ to 1024 but only 0.83 from 1024 to 2048, and the single-level lookup shows the same

Table 8: Online performance on Foursquare-NYC (800 users, BiTULER, mean over three seeds) as the total location embedding dimension $d$ varies, with the recurrent hidden size held fixed, comparing MiLES against the single-level POI lookup and a Fourier-feature encoder. Per-seed standard deviations are below 0.15 in every cell. The best value per column and metric is in bold.

| Metric | Embedding | $d = 128$ | $d = 256$ | $d = 512$ | $d = 1024$ | $d = 2048$ |
|--------|-----------|-----------|-----------|-----------|------------|------------|
| | MiLES | **53.49** | **57.52** | **60.11** | **61.60** | **62.43** |
| Acc@1 | POI-only | 44.93 | 51.91 | 56.90 | 60.11 | 61.69 |
| | Fourier | 48.57 | 53.82 | 57.83 | 60.24 | 61.27 |
| | MiLES | **61.77** | **66.14** | **69.07** | **70.77** | **71.69** |
| Acc@5 | POI-only | 52.49 | 58.98 | 63.79 | 67.23 | 68.78 |
| | Fourier | 59.76 | 64.63 | 68.29 | 70.42 | 71.27 |
| | MiLES | **50.01** | **54.73** | **57.79** | **59.54** | **60.45** |
| F1 | POI-only | 39.84 | 47.80 | 53.89 | 57.82 | 59.72 |
| | Fourier | 44.77 | 50.79 | 55.34 | 58.11 | 59.26 |

Table 9: Performance of BiTULER on Foursquare-NYC by diversity in the shared MiLES grid cells visited by each trajectory. Diversity is normalized entropy over either POI categories or user labels. Metrics are reported in percent.

| Bin | Embedding | Category | | | User | | |
|-----|-----------|----------|-------|----------|-------|-------|----------|
| | | Acc@1 | Acc@5 | Macro F1 | Acc@1 | Acc@5 | Macro F1 |
| Low | POI-only | 73.39 | 78.81 | 46.82 | 80.32 | 84.36 | 47.93 |
| Low | MiLES | 74.89 | 80.84 | 48.69 | 82.90 | 87.65 | 50.44 |
| Medium | POI-only | 61.47 | 69.09 | 51.72 | 58.92 | 66.73 | 47.97 |
| Medium | MiLES | 62.90 | 72.82 | 54.00 | 60.92 | 71.57 | 50.67 |
| High | POI-only | 45.49 | 53.70 | 37.06 | 41.12 | 50.51 | 27.82 |
| High | MiLES | 47.02 | 58.68 | 39.91 | 41.00 | 53.12 | 28.66 |

pattern. The Fourier encoder, which realizes the *grid* variant of Space2Vec's coordinate encoding without its full-dataset pre-training, is likewise not a degenerate baseline. Because the total dimension is fixed, it devotes only half of $d$ to POI embeddings, yet at all but the largest budget its coordinate features more than compensate for the reduced POI capacity, outperforming the POI-only lookup that spends the entire budget on POIs. MiLES nonetheless leads at every budget, indicating that its structured multi-level sharing exploits spatial regularities more effectively than a continuous coordinate encoding. We benchmark against the single-level POI lookup throughout because it remains the embedding adopted by most TUL models. The reversal on GeoLife appears to stem from how location is represented on each dataset. On the POI-based Foursquare datasets, the finest embedding level indexes individual POIs, giving each venue a precise and distinct representation. GeoLife has no POIs, so its finest level is a grid cell, and neighboring points then receive an identical embedding even at high grid resolution, leaving the model unable to distinguish, for example, the road from the sidewalk of the same street. A continuous coordinate encoder instead assigns each point a distinct code and preserves this fine spatial detail, which appears more valuable on GeoLife's dense, continuous trajectories. Consistent with this, on the continuous-target destination-prediction task (Section 7) the same Fourier encoder instead matches MiLES and outperforms the POI-only lookup on every dataset and metric.

We also evaluated whether MiLES suffers negative transfer when shared grid cells contain heterogeneous behavior. For each shared MiLES grid level on Foursquare-NYC, we computed normalized entropy over both POI categories and user labels. Each trajectory was assigned the average diversity of the shared cells visited by its check-ins, averaged over grid-based embedding levels 1–3, and trajectories were split into tertiles.

Table 9 shows that higher diversity makes the task harder for both embeddings, as expected. Category diversity directly captures semantic heterogeneity among nearby POIs, while user diversity captures whether shared cells mix behavior from many different users. MiLES improves all metrics across category-diversity bins, including the high-diversity bin. This suggests that category diversity within a cell does not necessarily imply harmful sharing, which could be due to the fact that a neighborhood such as a shopping district can contain diverse POI categories while still having a coherent local function that shared spatial embeddings can capture. For user diversity, MiLES retains higher top-5 accuracy and Macro F1 across all bins, but its top-1 accuracy is marginally lower in the high-diversity bin, differing from the POI-only baseline by about 0.1%. We do not interpret this as strong evidence of active negative transfer. Because our experiments keep the total embedding dimension fixed, MiLES necessarily allocates fewer dimensions to the POI-level embedding, and Table 8 shows that reducing POI-level capacity alone degrades performance. The high user-diversity bin therefore appears to be a case where the shared levels provide less benefit for exact user identification, rather than a clear case where they are harmful. Nevertheless, diverse cells remain a natural boundary case, and future work could explore adaptive sharing mechanisms such as dynamically splitting high-diversity cells.

**MiLES Across Training Paradigms** To disentangle the contribution of MiLES's representational capacity from its interaction with the online learning setting, we compare the performance gains of MiLES over the baseline in both batch and online evaluation modes. Hyperparameters for this comparison were tuned separately from the main online experiments. We split the first 5,000 trajectories from Foursquare-TKY into training and test sets and selected the values providing the best accuracy on the test set after 20 training epochs. These batch-tuned hyperparameters were then used for both the batch and online evaluation modes. Note that the batch evaluation uses a random 80/20 train-test split to eliminate the effect of distributional shift and isolate MiLES's representational contribution from its interaction with non-stationarity.

As shown in table 10, MiLES improves performance in both settings, confirming that the multi-level architecture provides a general representational benefit. However, the gains are consistently larger in the online setting, most notably on GeoLife (+22.10% vs. +4.81% top-1 accuracy). We attribute this to the greater severity of gradient sparsity in the online regime, where each trajectory is observed only once and POI embeddings receive far fewer updates. That MiLES yields consistent improvements despite the batch-tuned hyperparameters differing in most of their values from the online-tuned ones provides additional evidence for its robustness to hyperparameter choices.

To investigate the middle ground between batch and online learning, we evaluate BiTULER under periodic batch retraining with intervals of 2,500 and 5,000 trajectories. At each interval, the model is reinitialized and trained on all accumulated data for 10 epochs. Between retraining steps, the model is frozen and predictions are collected prequentially as trajectories arrive, ensuring comparability with the online evaluation protocol. As in the batch comparison above, all paradigms in this experiment use the same batch-tuned hyperparameters, so that the differences reflect the training paradigm rather than the tuning protocol.

As Figure 8 and table 11 show, online learning consistently outperforms periodic retraining across all datasets and embedding types. Because this experiment uses the batch-tuned configuration for every paradigm, the online rows in Table 11 differ slightly from the online-tuned main results in Table 2, which report our best-case online performance. Online learning also requires fewer computations. Retraining every 2,500 trajectories on Foursquare-NYC, for example, involves approximately 120 times more forward and backward passes than online learning, since each retraining step processes all accumulated data for multiple epochs.

Table 10: Relative performance gain [%] of MiLES over the baseline in batch and online evaluation. We report relative rather than absolute gains to facilitate comparison across the two settings, which differ in baseline performance levels. Hyperparameters were tuned for the batch setting and used for both modes.

| Dataset | Foursquare-NYC | | | Foursquare-TKY | | | GeoLife | | |
|---|---|---|---|---|---|---|---|---|---|
| Scenario | Acc@1 | Acc@5 | Macro F1 | Acc@1 | Acc@5 | Macro F1 | Acc@1 | Acc@5 | Macro F1 |
| Batch | +4.24 | +6.41 | +3.84 | +3.97 | +4.80 | +3.93 | +4.81 | +2.52 | +8.10 |
| Online | +4.32 | +8.04 | +5.31 | +4.44 | +6.62 | +4.69 | +22.10 | +10.03 | +21.57 |

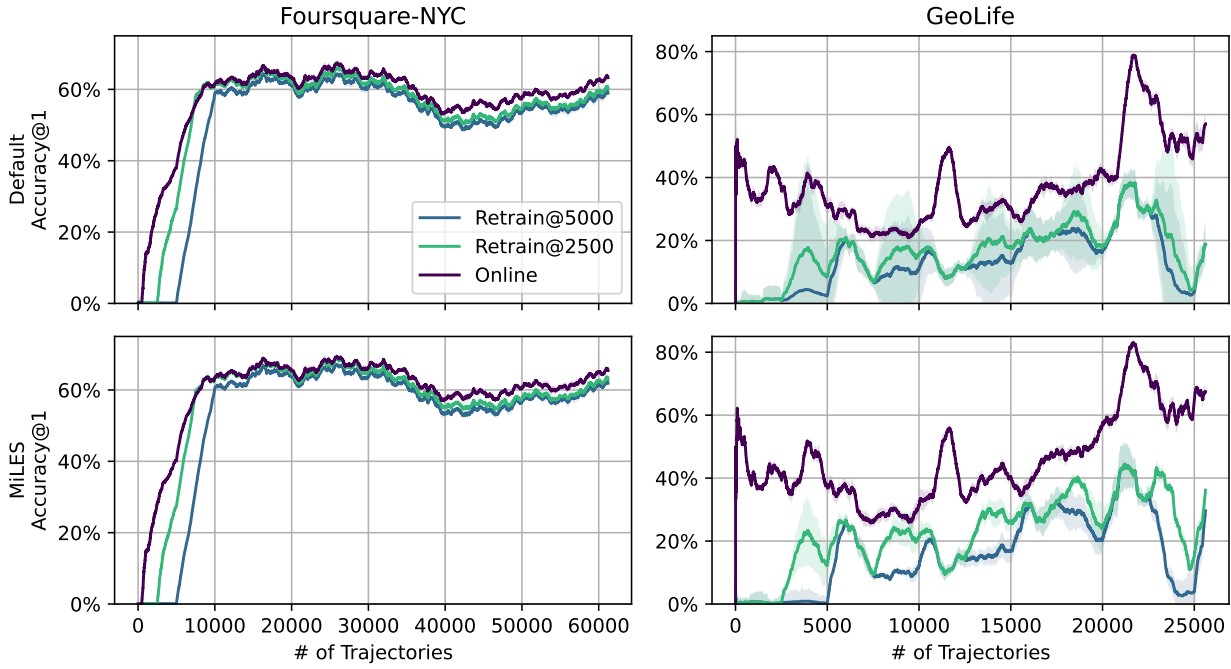

Figure 8: Rolling top-1 accuracy for BiTULER under periodic retraining (every 5,000 and 2,500 trajectories) and online learning, for the default embedding (top) and MiLES (bottom). Sliding window sizes are 5,000 (Foursquare-NYC) and 1,000 (GeoLife). Shaded areas indicate the $2\sigma$ interval over 5 random seeds.

The advantage is most pronounced on GeoLife, where periodic retraining achieves only 13.94–16.95% top-1 accuracy compared to 37.82% for online learning. We attribute this to two factors: the model cannot adapt between retraining intervals, and retraining on all accumulated data increasingly biases the model toward earlier distributional patterns. This effect is particularly severe on GeoLife, where user composition shifts over time (Figure 14) and consistent with the observation that the recency-prioritizing FIFO replay buffer outperforms the class-balanced one (Table 6).

Taken together, these results suggest that online learning is well-suited for TUL applications that require real-time predictions on streaming data. MiLES improves performance across all training paradigms. Its relative gains scale with the degree of distributional mismatch between the training and evaluation data. The gains are smallest in batch learning, larger in online learning, and largest under periodic retraining, where the model's knowledge lags behind the current stream.

Table 11: Top-1 accuracy (%) of BiTULER across training paradigms. Periodic retraining reinitializes the model at fixed intervals and trains on all accumulated data for 10 epochs. All rows use the same batch-tuned hyperparameters. $\Delta$ MiLES denotes the change in accuracy when using MiLES instead of the default embedding.

| Emb. | Dataset Interval | Foursquare-NYC | | | Foursquare-TKY | | | GeoLife | | |
|---|---|---|---|---|---|---|---|---|---|---|
| | | Acc@1 | Acc@5 | F1 | Acc@1 | Acc@5 | F1 | Acc@1 | Acc@5 | F1 |
| Default | 5000 | 52.72 | 59.45 | 51.35 | 55.30 | 65.56 | 52.85 | 13.94 | 27.24 | 3.63 |
| | 2500 | 56.23 | 63.20 | 55.22 | 58.34 | 69.00 | 56.23 | 16.95 | 33.03 | 5.02 |
| | Online | **59.03** | **65.85** | **56.38** | **59.03** | **70.71** | **56.25** | **37.82** | **70.41** | **25.78** |
| MiLES | 5000 | 55.45 | 64.47 | 53.38 | 58.13 | 69.87 | 55.29 | 16.86 | 30.64 | 4.36 |
| | 2500 | 59.01 | 68.31 | 57.29 | 61.19 | 73.36 | 58.66 | 23.55 | 42.16 | 7.40 |
| | Online | **61.58** | **71.15** | **59.37** | **61.65** | **75.38** | **58.89** | **46.17** | **77.46** | **31.34** |

# 7 Generalization to Destination Prediction

To demonstrate the general applicability of MiLES beyond Trajectory-User Linking, we conducted additional experiments on the task of destination prediction, which has considerable practical importance for applications such as personal navigation systems or ride sharing platforms (Endo et al., 2017). The objective of this task is to predict the coordinates of the final check-in $c_n$, of a trajectory given an initial sub-trajectory of check-ins $[c_0, c_1, \ldots, c_k]$, where the lengths of the complete trajectory $n$ and the input sequence $k$ may vary.

We emphasize that we use destination prediction solely to assess the generality of MiLES, not to establish new state-of-the-art results for this task.

## 7.1 Experimental Setup

Our experimental setup is based on the ECML/PKDD 2015 discovery challenge on taxi destination prediction (de Brébisson et al., 2015). For each dataset, we generated a stream of 50,000 pairs of partial trajectories and their final destinations by sampling sub-trajectories at random points. For the GeoLife dataset, we first filtered out check-ins outside the greater Beijing area and subsampled the remaining data at 10-minute intervals.

We framed the task as a regression problem. To ensure that the Euclidean distance accurately reflect real-world distances, we projected the destination coordinates $l_n = [\text{longitude}_n, \text{latitude}_n]$ into a local planar coordinate system. All models were then trained to predict these projected coordinates, allowing the prediction error to be calculated as the straight-line distance.

In terms of model architecture, we also employed a similar approach to that used in de Brébisson et al. (2015): We encode partial trajectories, as well as the weekday and user-ID associated with the respective trajectory. For the latter, we use embedding vectors with 256 dimensions each. All encoded features are then fed to a fully connected layer, producing the final prediction.

Previous work used a variety of neural architectures like multi-layer perceptrons (MLPs) (de Brébisson et al., 2015), convolutional neural networks (CNNs) (Lv et al., 2018) or LSTMs (Endo et al., 2017; Ebel et al., 2020) for the purpose of destination prediction. Based on this, we used either a bidirectional LSTM (BiLSTM), a convolutional neural network (CNN), a transformer (Transformer) or an MLP to encode the partial trajectories For the latter, we computed the average of all check-in embeddings to aggregate the partial trajectory.

We tune the hyperparameters of all models and techniques using the initial 5000 partial trajectories of the 400 user Foursquare-TKY dataset.

## 7.2 Results

We evaluated all models using either a default single-level POI lookup embedding or the proposed MiLES embedding. Table 12 presents the mean, median (P50), and 95th percentile (P95) of the prediction errors, averaged over five prequential evaluation runs.

Among the baseline models, the CNN performed best on the Foursquare datasets, while the BiLSTM was superior on GeoLife. The Transformer model yielded the worst results across all datasets, which is likely attributable to the short average length of the partial trajectories (e.g., only five check-ins for Foursquare-NYC), limiting the effectiveness of its self-attention mechanism. Consistent with our TUL experiments, integrating MiLES improved destination prediction performance across most models, metrics, and datasets. The Foursquare datasets, which have fewer check-ins per trajectory, benefited most, with MiLES achieving reductions in mean error of up to 11%.

The performance gains were less pronounced for the CNN model, and on Foursquare-TKY its mean error even rose marginally by 0.05 km. At a fixed total dimension, MiLES trades POI-level capacity for coarser shared levels, which on its own reduces model performance considerably (Table 8). This is most pronounced for the CNN, whose kernels already smooth nearby POIs through their local receptive field, so the coarser shared levels are partially redundant and add little. Consequently, the shared levels no longer overcompensate for

Table 12: Mean, 50th-, and 95th percentile of distances between predicted and actual destinations in kilometers for models using a basic lookup embedding (Default) and performance gains when using our technique (Δ MiLES).

| Dataset | Foursquare-NYC | | | Foursquare-TKY | | | GeoLife | | |
|---|---|---|---|---|---|---|---|---|---|
| Model | Mean | P50 | P95 | Mean | P50 | P95 | Mean | P50 | P95 |
| Default | | | | | | | | | |
| BiLSTM | 7.21 | 5.37 | 20.03 | 6.97 | 5.60 | 17.68 | **5.96** | **4.29** | **16.90** |
| CNN | **6.51** | **4.81** | **18.28** | **6.57** | **5.21** | **16.86** | 6.33 | 4.56 | 17.87 |
| MLP | 7.80 | 6.14 | 20.08 | 7.53 | 6.23 | 18.09 | 7.12 | 5.39 | 19.10 |
| Transformer | 8.90 | 7.26 | 21.81 | 8.11 | 6.93 | 18.58 | 7.54 | 5.81 | 19.97 |
| Δ MiLES | | | | | | | | | |
| BiLSTM | -0.72 | -0.65 | -1.56 | -0.24 | -0.29 | -0.24 | -0.09 | -0.07 | -0.34 |
| CNN | -0.13 | -0.18 | -0.11 | 0.05 | 0.04 | 0.13 | -0.20 | -0.20 | -0.28 |
| MLP | -0.55 | -0.52 | -0.99 | -0.16 | -0.18 | -0.21 | -0.16 | -0.17 | -0.25 |
| Transformer | -1.01 | -0.97 | -1.89 | -0.48 | -0.51 | -0.59 | -0.09 | -0.09 | -0.17 |

Table 13: Mean, 50th-, and 95th percentile of distances between predicted and actual destinations in kilometers for different embedding- and experience replay methods, averaged across all models shown in Table 12. [†]The default configuration uses POI-based lookup embeddings without replay.

| Dataset | Foursquare-NYC | | | Foursquare-TKY | | | GeoLife | | |
|---|---|---|---|---|---|---|---|---|---|
| Method | Mean | P50 | P95 | Mean | P50 | P95 | Mean | P50 | P95 |
| Default[†] | 7.61 | 5.90 | 20.05 | 7.47 | 6.14 | 18.22 | 6.74 | 5.01 | 18.46 |
| Embedding | | | | | | | | | |
| Linear | 7.13 | 5.56 | **18.63** | 7.20 | 5.92 | **17.55** | 6.83 | 5.12 | 18.53 |
| Fourier | **6.98** | **5.30** | 18.90 | 7.09 | **5.75** | 17.63 | **6.57** | **4.87** | **18.01** |
| MiLES | 7.00 | 5.32 | 18.91 | **7.09** | 5.76 | 17.57 | 6.60 | 4.88 | 18.20 |
| Replay | | | | | | | | | |
| FIFO | 7.39 | 5.52 | 20.35 | 7.30 | 5.82 | 18.64 | 6.37 | 4.50 | 18.33 |
| Random | 7.39 | 5.50 | 20.41 | 7.31 | 5.82 | 18.62 | 6.31 | 4.43 | 18.27 |
| FIFO+MiLES | **6.82** | **4.97** | **19.35** | **7.10** | **5.56** | **18.45** | **6.09** | **4.26** | **17.67** |

the lost POI-level dimensions. Even in this case, however, MiLES retains its parameter-efficiency advantage over the default embedding, so the marginal increase of 0.05 km is incurred at a smaller embedding size rather than as an outright regression. We also note that MiLES uses a single shared configuration tuned for the default embedding rather than one adapted per backbone, so a CNN-specific configuration could plausibly close this small gap.

We further compared MiLES against alternative embedding and experience replay techniques, with results averaged over all models reported in Table 13. While linear embeddings achieved a lower 95th percentile error on the Foursquare datasets, MiLES achieved the best mean error on Foursquare-TKY and otherwise closely matched a properly tuned Fourier embedding across datasets, echoing the same pattern we found for the Fourier embedding on GeoLife in the trajectory user linking task. As in that setting, MiLES reaches this level of accuracy from a discrete multi-level embedding rather than a continuous coordinate encoding, remaining competitive even where it no longer leads outright on error.

Experience replay strategies reduced mean and median errors. However, they did not improve the 95th percentile error, suggesting that while replay reinforces common travel patterns, it may not help with predicting less frequent or novel destinations that constitute the long-tail of the error distribution.

Notably, the combination of MiLES with a FIFO replay buffer (FIFO + MiLES) achieved the best results across nearly all metrics and datasets, once again demonstrating that MiLES can be used to complement and enhance other online learning strategies.

## 8 Conclusion

In this paper, we investigated online learning for Trajectory-User Linking on streaming trajectory data. To the best of our knowledge, this work presents the first systematic study of TUL in an online learning setting. We introduced Multi-Level Spatial Embedding Sharing (MiLES), an embedding method that balances representational density and specificity through a weighted, multi-scale architecture, with particular benefits in online learning. Our experimental evaluation provided three key findings. First, in the context of TUL, MiLES consistently enhanced the performance and efficiency of several state-of-the-art models across batch, periodic retraining, and online learning paradigms. MiLES thus provides an effective alternative to the standard POI-only lookup embedding on which most TUL models rely. On denser, continuous-GPS data such as GeoLife, where locations are not tied to discrete POIs, a continuous coordinate encoder proves more effective, with MiLES remaining the strongest of the discrete lookup-based embeddings. Second, online learning consistently outperformed periodic batch retraining at lower computational cost, with the gap being particularly large on datasets exhibiting distributional shift, where retraining on accumulated historical data increasingly biases the model. Third, we demonstrated the generalizability of MiLES on the task of destination prediction, where it again improved prediction accuracy across multiple backbone architectures. Overall, these findings establish MiLES as a broadly applicable embedding technique for data-scarce mobility settings, and support online learning as a natural paradigm for TUL applications requiring real-time predictions on streaming data. Our controlled drift experiment shows that MiLES maintains higher performance when previously unobserved locations enter the stream, and the rolling accuracy analysis confirms that this advantage persists under naturally occurring distributional shifts in unmodified datasets.

## 9 Limitations

Although MiLES shows consistent improvements in performance and efficiency for trajectory-user linking and destination prediction, its core inductive bias that spatially proximate locations share useful structure may not apply equally in all settings. Our cell-diversity analysis (Table 9) offers a nuanced picture of when this matters. Semantic heterogeneity among nearby POIs, such as a transport hub adjacent to a residential area, does not in itself lead to harmful sharing. However, cells that mix many different users' behavior are a boundary case where shared grid-level embeddings provide less benefit for exact user identification. In such cases, a single-level, POI-specific embedding that devotes full representational capacity to distinguishing individual locations may be preferable, particularly if there is enough training data to overcome gradient sparsity without sharing. More generally, we believe MiLES is most beneficial when two conditions are met: when spatial proximity is at least partially informative and when the per-location training signal is sparse. These conditions are typical of the online TUL setting but not unique to it. Relatedly, while we observe consistent benefits across recurrent, attention-based, and convolutional backbones, our evidence does not let us claim that MiLES improves every possible architecture, with the convolutional model on the dense Foursquare-TKY dataset marking the one scenario in our study where its spatial sharing adds no benefit.

Furthermore, although MiLES reduces the rate of parameter growth compared to baseline methods (as shown in Table 5), it relies on the dynamic expansion of embedding tables whenever new grid cells or POIs are encountered. In our experiments, pre-allocation sidesteps this issue because the full spatial extent is known, but in a true unbounded stream, embedding tables would grow monotonically with the number of distinct grid cells observed. This means that MiLES may eventually exhaust available memory for a long-running stream over a large or expanding spatial domain. Future work could address this by investigating fixed-budget mechanisms such as embedding pruning or fixed-size hash-based tables, for example the feature hashing of Hu et al. (2022), which is directly compatible with MiLES's per-level tables.

Our evaluation follows the standard supervised online learning protocol, in which the true user label is assumed to be available immediately following each prediction. In practice, however, feedback may be

delayed or incomplete, and the behavior of the model under such conditions remains unexplored. While our comparison against periodic batch retraining (Section 6) shows a clear advantage for online learning, we evaluate only full retraining on all accumulated data. Alternative strategies such as windowed retraining or weighted sampling of recent data could mitigate the staleness effects we observe and may narrow the gap. Similarly, our controlled drift experiment and rolling accuracy analysis provide converging evidence that MiLES is robust to the types of distributional shift present in our datasets, but the extent to which this generalizes to different types and magnitudes of drift remains an open question. Promising directions for future work include investigating alternative retraining strategies, the effectiveness of MiLES under semi-supervised feedback, and its robustness to a broader range of non-stationarity scenarios.

**Broader Impact Statement**

By improving the performance of mobility data mining models, the methods presented in this paper have important broader impacts. While they could enable societal benefits in areas like urban planning and logistics, they also pose privacy risks if misused for surveillance or unwanted tracking. More capable models increase this risk regardless of training paradigm, as higher accuracy enables more reliable re-identification of individuals. Consequently, any real-world deployment should incorporate robust privacy-preserving mechanisms, such as differential privacy (Mir et al., 2013; Yin et al., 2018), or anonymization as a prerequisite for ethical application. At the same time, the online setting central to our work allows trajectory data to be discarded immediately after each model update, eliminating the need to retain sensitive movement records for training purposes.

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

# A   Experimental Details and Reproducibility

## A.1   MiLES Embedding Construction

Algorithm 1 summarizes MiLES's embedding construction for a single check-in, including the online dynamic expansion of the per-level tables described in Section 4.

---

**Algorithm 1:** MiLES: Multi-level Spatial Embedding Sharing

---

**Input:** Check-in $c$, Embedding dims $d_l$
**Parameters:** Embedding matrices $\{Z_l\}$, Level weights $w$
**Output:** Embedding vector $\mathbf{g}$

1   $l \leftarrow c.$coordinates;
2   $\mathcal{E} \leftarrow [\,]$ ;                      // Initialize list for level-specific embeddings.
3   **for** $l \leftarrow 0$ **to** $l^{(\max)}$ **do**
4      **if** $l = 0$ **and** $c$ *contains POI identifier* $p$ **then**
5          $h_l \leftarrow p$;
6      **else**
7          $h_l \leftarrow \text{GetHexGridIndex}(l, \text{level} = l)$;
8      **if** $h_l \geq \text{rows}(Z_l)$ **then**
9          $\mathbf{r}_{\text{new}} \leftarrow \text{InitializeVector}(d_l)$;
10        $Z_l \leftarrow \text{AppendRow}(Z_l, \mathbf{r}_{\text{new}})$;             // Expand embedding table.
11      $\mathbf{z} \leftarrow Z_l[h_l]$;
12      $\mathcal{E} \leftarrow \mathcal{E} \cup \{\mathbf{z} \cdot w_l\}$ ;            // Apply level weight and append result.
13   $\mathbf{g} \leftarrow \text{Concat}(\mathcal{E})$;               // Concatenate results of all levels.
14   **return** $\mathbf{g}$;

---

## A.2   Hyperparameter Values

As described in the main paper, all models and techniques were tuned using the first 5,000 trajectories of the 400-user Foursquare-TKY stream. This protocol was applied identically to all methods to ensure a fair comparison. Table 14 lists the resulting hyperparameter values used in our experiments. For MiLES, $l^{(\max)} = 4$ corresponds to one POI-level embedding and three grid-based sharing levels on Foursquare datasets, and four grid-based levels on GeoLife, which lacks POI information.

Table 14: Tested hyperparameter values for the evaluated models and methods. The selected values are underlined. If different values were selected depending on the used model, multiple are underlined.

| Hyperparameter | Symbol | Tested Values |
|---|---|---|
| Full embedding dim. | $d$ | $\{512, \underline{1024}\}$ |
| Embedding levels | $l^{(\max)}$ | $\{3, \underline{4}, 5\}$ |
| Base level grid rows | $h_1^{(\max)}$ | $\{100, \underline{200}, \dots, 500\}$ |
| Level $l$ grid rows | $h_l^{(\max)}$ | $\{\underline{2}, 3, 4\}^{-l} \times h_1^{(\max)}$ |
| Embedding dim. decay | $\alpha$ | $\{1, \underline{2}\}$ |
| Fourier feature scale | — | $\{1, 2, \underline{4} \dots, 16\} \times 10^3$ |
| Replay samples/update | — | $\{\underline{1}, 2, 4\}$ |
| Optimizer | — | Adam |
| Learning rate | — | $\{.5, \underline{1}, \underline{2}, \underline{4}, 8\} \times 10^{-4}$ |
| Hidden layers | — | $\{\underline{1}, \underline{2}\}$ |
| Hidden units | — | $\{512, \underline{1024}\}$ |

**Sensitivity to the Tuning Window**    Since hyperparameters are selected on the earliest 5,000 trajectories of the stream, we test whether this specific choice of tuning window affects our conclusions. We repeated the MiLES hyperparameter sweep for BiTULER on Foursquare-TKY using a different, non-overlapping 5,000-trajectory window drawn from later in the same stream (trajectories 20,000–25,000), keeping all other protocol details fixed. The resulting best configuration differs from the one selected on the original window in three of four hyperparameters (Table 15). We then evaluated both configurations on a common held-out segment unseen by either tuning run (all trajectories from index 25,000 onward). The two configurations perform within 1 pp. of each other on every metric, with the configuration tuned on the later window consistently, if only marginally, ahead. This indicates that, while the optimal hyperparameters can shift across tuning windows, downstream performance is not sensitive to this choice, and if anything our early-window protocol is a slightly conservative choice rather than one that is tailored to inflate MiLES's reported gains. This is consistent with Figure 5, which shows that performance is robust to the base grid resolution once it is sufficiently high rather than sharply peaked at a single value. Notably, the four-level configuration there also achieves its highest top-1 accuracy at 400 base-level grid rows, the same value selected on the later tuning window.

Table 15: Best MiLES configuration selected on the original tuning window (trajectories 0–5,000) versus a later, non-overlapping window (trajectories 20,000–25,000) of the same Foursquare-TKY stream, and resulting top-1, top-5 accuracy and macro F1 [%] when each configuration is trained from scratch over the full stream and evaluated on trajectories 25,000 onward, averaged over 5 seeds.

| Tuning window | $l^{(\max)}$ | $h_1^{(\max)}$ | Grow factor | Accuracy@1 | Accuracy@5 | Macro F1 |
|---|---|---|---|---|---|---|
| 0–5,000 (original) | 4 | 200 | 2 | $73.91 \pm 0.17$ | $85.92 \pm 0.04$ | $69.44 \pm 0.25$ |
| 20,000–25,000 | 5 | 400 | 4 | $74.25 \pm 0.20$ | $86.66 \pm 0.07$ | $69.93 \pm 0.42$ |

## A.3   Computing Infrastructure

- GPU: Nvidia RTX 3090

- CPU: Intel Core i5-9600K

- System Memory: 32 GB DDR4@2133MT/s

- OS: Ubuntu 22.04 LTS

- CUDA: 12.5

- Python: 3.12

- PyTorch: 2.3.1

## A.4   Embedding Parameter Counts

Table 16 reports the per-level embedding-parameter breakdown underlying the totals in Table 5. At a fixed total embedding dimension of 1024, MiLES distributes the dimensions across levels with a geometrically decaying schedule, so the POI level receives 547 dimensions and the three coarser grid levels receive 273, 136 and 68, respectively. Because the coarse grids contain far fewer distinct cells than there are POIs, the resulting tables are small, and the overall embedding is reduced by roughly 38% on the Foursquare datasets relative to the single-level default.

Table 16: Per-level embedding-parameter breakdown for MiLES (Table 5), using the main-experiment configuration (four levels, growth factor 2, total embedding dimension 1024). *Locations* is the number of distinct indices at that level (POIs at level 0 for the Foursquare datasets, grid cells otherwise). The default embedding assigns the full 1024 dimensions to a single POI level. GeoLife has no POI information, so level 0 is the finest grid.

| Dataset | Level | Locations | Dim | Parameters |
|---------|-------|-----------|-----|------------|
| Foursquare-NYC | 0 (POI) | 34,384 | 547 | 18,808,048 |
| | 1 (Grid 1) | 8,869 | 273 | 2,421,237 |
| | 2 (Grid 2) | 4,564 | 136 | 620,704 |
| | 3 (Grid 3) | 1,789 | 68 | 121,652 |
| | MiLES total | | 1024 | 21,971,641 |
| | Default | 34,384 | 1024 | 35,209,216 |
| Foursquare-TKY | 0 (POI) | 38,213 | 547 | 20,902,511 |
| | 1 (Grid 1) | 9,589 | 273 | 2,617,797 |
| | 2 (Grid 2) | 4,926 | 136 | 669,936 |
| | 3 (Grid 3) | 1,950 | 68 | 132,600 |
| | MiLES total | | 1024 | 24,322,844 |
| | Default | 38,213 | 1024 | 39,130,112 |
| GeoLife | 0 (Grid 1) | 4,627 | 547 | 2,530,969 |
| | 1 (Grid 2) | 2,163 | 273 | 590,499 |
| | 2 (Grid 3) | 976 | 136 | 132,736 |
| | 3 (Grid 4) | 442 | 68 | 30,056 |
| | MiLES total | | 1024 | 3,284,260 |
| | Default | 4,627 | 1024 | 4,738,048 |

## A.5 Full Results for All Dataset Variants

The results for all evaluated TUL approaches as well as the dataset-variants with lower user counts are displayed in Table 17. As expected, all models achieved better performance on the lower user-count datasets. In terms of individual models, BiTULER remains the overall most performant model independent of the user count, except for the GeoLife dataset where the GRU-variant of TULER and DeepTUL yielded better results.

The bottom section of Table 17 shows consistent performance gains when substituting the original embedding techniques of the evaluated models with MiLES for both high- and low user-count datasets. The performance benefits at lower user-counts are even higher compared to the larger dataset variants with top-1 accuracy gains of up to 9.79%. The larger benefit for data with fewer users likely stems from the fact that the lower user-count causes the individual users to be more easily identified based on the higher-level embeddings of MiLES.

Table 18 reports the variance of the $\Delta$ MiLES effects in Table 2, computed by pairing each MiLES run with the corresponding default-embedding run for the same model, dataset variant and seed before taking the difference. The improvement estimates are stable across seeds, with a standard deviation of at most 0.78 pp. (90th percentile 0.38) across all model, dataset and metric combinations. The largest standard deviation occurs for MainTUL's macro F1 gain on GeoLife, where the mean improvement is still 5.50 pp., about seven times the standard deviation. MiLES improves over the default embedding in all 18 model $\times$ dataset settings on every metric. Per-model paired significance tests are reported in Table 19.

A per-model breakdown of this comparison is given in Table 19. For every model, dataset and metric, a one-sided Wilcoxon signed-rank test pairing the full MiLES and default runs over both user-count variants and five seeds (N = 10 paired observations) rejects the null hypothesis that MiLES does not improve over the default embedding at the minimum attainable level ($p \approx 0.001$), as MiLES outperforms the default in every paired run.

Table 17: Top-1 accuracy, top-5 accuracy and macro F1 [%], as well as their diffences when using MiLES instead of the default embeddings, for all TUL models on all dataset variants.

| | Dataset | Foursquare-NYC | | | Foursquare-TKY | | | GeoLife | | |
|---|---|---|---|---|---|---|---|---|---|---|
| # Users | Model | Acc@1 | Acc@5 | F1 | Acc@1 | Acc@5 | F1 | Acc@1 | Acc@5 | F1 |
| | | | | | Default | | | | | |
| Low | BiTULER | **69.95** | **75.85** | **68.59** | **73.29** | **84.06** | **72.03** | **39.98** | 74.16 | 32.95 |
| | TULVAE | 69.77 | 75.72 | 68.37 | 66.86 | 77.42 | 64.05 | 39.40 | 73.82 | 31.57 |
| | DeepTUL | 68.93 | 74.77 | 67.62 | 71.65 | 82.47 | 70.42 | 38.44 | 75.15 | **33.42** |
| | MainTUL | 66.28 | 72.41 | 64.86 | 70.69 | 82.30 | 69.28 | 35.89 | 73.02 | 27.87 |
| | T3S | 63.84 | 70.42 | 61.84 | 66.77 | 79.41 | 64.89 | 37.79 | 74.68 | 29.06 |
| | TULHOR | 64.60 | 71.08 | 62.69 | 67.21 | 79.57 | 65.36 | 37.22 | **75.43** | 30.87 |
| High | BiTULER | **60.12** | **67.20** | **57.83** | **63.16** | **74.91** | **61.06** | **37.56** | 70.85 | 26.69 |
| | TULVAE | 59.79 | 66.77 | 57.32 | 55.82 | 66.23 | 51.68 | 37.08 | 70.45 | 25.25 |
| | DeepTUL | 58.72 | 65.48 | 56.60 | 61.17 | 72.49 | 59.10 | 36.32 | **72.64** | **29.82** |
| | MainTUL | 55.67 | 62.61 | 53.01 | 59.53 | 71.89 | 57.09 | 34.00 | 70.26 | 21.76 |
| | T3S | 52.98 | 60.28 | 49.50 | 56.41 | 69.26 | 53.24 | 35.25 | 71.11 | 21.52 |
| | TULHOR | 53.85 | 61.13 | 50.40 | 56.61 | 69.61 | 53.45 | 34.65 | 72.46 | 24.92 |
| | | | | | Δ MiLES | | | | | |
| Low | BiTULER | +2.15 | +3.99 | +2.43 | +1.38 | +2.24 | +1.44 | +8.97 | +6.80 | +8.40 |
| | TULVAE | +2.22 | +4.02 | +2.51 | +2.88 | +3.80 | +3.40 | +8.96 | +6.82 | +8.70 |
| | DeepTUL | +1.67 | +3.54 | +1.98 | +0.80 | +1.95 | +0.76 | +8.93 | +6.15 | +8.55 |
| | MainTUL | +2.17 | +4.67 | +2.29 | +1.25 | +2.54 | +1.08 | +9.11 | +6.65 | +8.68 |
| | T3S | +1.83 | +3.09 | +2.07 | +1.77 | +2.20 | +1.85 | +7.09 | +4.39 | +7.00 |
| | TULHOR | +1.81 | +3.00 | +2.06 | +1.64 | +2.10 | +1.65 | +8.21 | +4.71 | +7.51 |
| High | BiTULER | +1.49 | +3.58 | +1.71 | +1.40 | +2.85 | +1.30 | +8.72 | +6.98 | +6.26 |
| | TULVAE | +1.79 | +3.73 | +2.04 | +3.07 | +4.43 | +3.34 | +8.01 | +6.57 | +4.74 |
| | DeepTUL | +1.06 | +3.04 | +1.26 | +0.84 | +2.37 | +0.77 | +8.59 | +6.31 | +5.97 |
| | MainTUL | +1.44 | +4.33 | +1.52 | +1.62 | +3.45 | +1.45 | +8.19 | +6.18 | +5.50 |
| | T3S | +1.71 | +3.13 | +2.10 | +1.70 | +2.79 | +1.91 | +6.98 | +4.80 | +5.18 |
| | TULHOR | +1.71 | +3.16 | +2.11 | +1.85 | +2.80 | +2.03 | +8.20 | +5.03 | +5.68 |

Table 18: Mean $\pm$ standard deviation over five paired seeds for the Δ MiLES portion of Table 2. Each paired observation is computed as MiLES minus the default embedding for the same model, dataset, user-count variant and seed. Across all 54 model $\times$ dataset $\times$ metric improvements, the standard deviation is at most 0.78 pp. (90th percentile 0.38).

| Dataset | Foursquare-NYC | | | Foursquare-TKY | | | GeoLife | | |
|---|---|---|---|---|---|---|---|---|---|
| | Acc@1 | Acc@5 | F1 | Acc@1 | Acc@5 | F1 | Acc@1 | Acc@5 | F1 |
| BiTULER | $1.49\pm0.03$ | $3.58\pm0.04$ | $1.71\pm0.05$ | $1.40\pm0.08$ | $2.85\pm0.11$ | $1.30\pm0.05$ | $8.72\pm0.27$ | $6.98\pm0.33$ | $6.26\pm0.59$ |
| TULVAE | $1.79\pm0.24$ | $3.73\pm0.22$ | $2.04\pm0.30$ | $3.07\pm0.18$ | $4.43\pm0.10$ | $3.34\pm0.21$ | $8.01\pm0.09$ | $6.57\pm0.21$ | $4.74\pm0.16$ |
| DeepTUL | $1.06\pm0.07$ | $3.04\pm0.09$ | $1.26\pm0.09$ | $0.84\pm0.08$ | $2.37\pm0.10$ | $0.77\pm0.12$ | $8.59\pm0.24$ | $6.31\pm0.36$ | $5.97\pm0.38$ |
| MainTUL | $1.44\pm0.15$ | $4.33\pm0.06$ | $1.52\pm0.22$ | $1.62\pm0.15$ | $3.45\pm0.09$ | $1.45\pm0.17$ | $8.19\pm0.27$ | $6.18\pm0.36$ | $5.50\pm0.78$ |
| T3S | $1.71\pm0.07$ | $3.13\pm0.09$ | $2.10\pm0.09$ | $1.70\pm0.15$ | $2.79\pm0.16$ | $1.91\pm0.15$ | $6.98\pm0.39$ | $4.80\pm0.35$ | $5.18\pm0.49$ |
| TULHOR | $1.71\pm0.08$ | $3.16\pm0.09$ | $2.11\pm0.11$ | $1.85\pm0.17$ | $2.80\pm0.13$ | $2.03\pm0.13$ | $8.20\pm0.26$ | $5.03\pm0.25$ | $5.68\pm0.63$ |

Table 19: P-values of one-sided Wilcoxon signed-rank tests, with the null hypothesis that replacing each model's default embedding with MiLES does not improve the respective performance metric. Each paired observation is the metric difference (MiLES minus default) for a single prequential run, matched by user-count variant and seed. With two variants and five seeds per dataset, each test uses N = 10 paired observations. Because every paired run improves, all tests reach the minimum attainable value ($p \approx 0.001$): MiLES significantly improves every model on every dataset and metric. Significant values are highlighted as in Table 7.

| Dataset | Foursquare-NYC | | | Foursquare-TKY | | | GeoLife | | |
|---|---|---|---|---|---|---|---|---|---|
| Model | Acc@1 | Acc@5 | F1 | Acc@1 | Acc@5 | F1 | Acc@1 | Acc@5 | F1 |
| BiTULER | <0.001 | <0.001 | <0.001 | <0.001 | <0.001 | <0.001 | <0.001 | <0.001 | <0.001 |
| TULVAE | <0.001 | <0.001 | <0.001 | <0.001 | <0.001 | <0.001 | <0.001 | <0.001 | <0.001 |
| DeepTUL | <0.001 | <0.001 | <0.001 | <0.001 | <0.001 | <0.001 | <0.001 | <0.001 | <0.001 |
| MainTUL | <0.001 | <0.001 | <0.001 | <0.001 | <0.001 | <0.001 | <0.001 | <0.001 | <0.001 |
| T3S | <0.001 | <0.001 | <0.001 | <0.001 | <0.001 | <0.001 | <0.001 | <0.001 | <0.001 |
| TULHOR | <0.001 | <0.001 | <0.001 | <0.001 | <0.001 | <0.001 | <0.001 | <0.001 | <0.001 |

### A.6   Performance by Trajectory Overlap

To quantify ambiguity in the single-label TUL formulation, we grouped Foursquare-NYC trajectories by their similarity to previously observed trajectories from other users. For each trajectory, we represented every check-in as a tuple of its POI identifier and exact hour of day. We then computed the longest common subsequence between the current trajectory and all earlier trajectories from different users that shared at least one such tuple. The overlap score is the resulting LCS length divided by the length of the current trajectory. This gives a stream-compatible measure of how much of the currently evaluated trajectory could already have been attributed to another user based on the observable POI and hour sequence.

Table 20 reports the results for BiTULER on Foursquare-NYC with the POI-only embedding and with MiLES.

Table 20: Performance of BiTULER on Foursquare-NYC by prior different-user trajectory overlap. Overlap is measured by the LCS of hour-precise (POI, hour) tuples between the current trajectory and earlier trajectories from other users. # and % of Trajectories represents the average total and relative number of trajectories in the respective bin.

| Overlap bin | # of Traj. | % of Traj. | Embedding | Acc@1 | Acc@5 | Macro F1 |
|---|---|---|---|---|---|---|
| 0% | 37087 | 60.58 | POI-only | 62.85 | 67.41 | 57.86 |
| 0% | 37087 | 60.58 | MiLES | 64.86 | 71.77 | 59.68 |
| $\leq 50\%$ | 19576 | 31.98 | POI-only | 63.08 | 72.89 | 58.66 |
| $\leq 50\%$ | 19576 | 31.98 | MiLES | 64.38 | 75.84 | 60.58 |
| $> 50\%$ | 4555 | 7.44 | POI-only | 23.80 | 38.73 | 17.76 |
| $> 50\%$ | 4555 | 7.44 | MiLES | 23.57 | 40.16 | 17.98 |

The high-overlap bin is much harder than the other two bins for both embeddings. This supports the interpretation that a non-trivial share of errors is linked to ambiguous trajectories whose observable POI and hour patterns closely resemble trajectories previously generated by other users. MiLES improves performance in the no-overlap and partial-overlap bins, while the high-overlap bin remains difficult for both methods. This is consistent with the notion that spatial sharing can improve representation learning for more sparsely visited locations but cannot resolve cases where the same observed trajectory pattern is plausible for multiple users. We note that this ambiguity is inherent to the single-label TUL formulation itself, rather than being introduced by MiLES or by the online setting. Whenever two users produce the same observable trajectory, a single-label objective must attribute it to one of them.

These ambiguous cases point to two practical directions. First, part of the overlap stems from trajectories that are indistinguishable given only their POI and coarse hour sequence, so incorporating additional discriminative features, such as the day of the week or finer contextual signals, could reduce it by making otherwise identical patterns separable. Second, the presence of trajectories that are genuinely plausible for several users motivates evaluating and deploying TUL with top-$k$ predictions rather than a single label. In applications such as ride-sharing, recommending the $k$ most likely candidates is often sufficient, so a model trained under the standard single-label formulation can remain useful, which is reflected in the consistently higher top-5 accuracy we observe. For such applications, however, set-valued or multi-label formulations of TUL would be a more faithful problem statement, a promising direction previously explored for the related task of trajectory-based social-circle inference (Gao et al., 2018).

### A.7   Full Results for Embedding & Replay Methods

The full results comparing MiLES to other embedding techniques as well as general online learning approaches are shown in Table 21. On the POI-rich Foursquare datasets, MiLES outperforms all other embedding techniques across every model and metric. On GeoLife, whose dense GPS trajectories lack discrete check-ins, the continuous Fourier encoder is instead the strongest embedding across all models, while MiLES remains the best of the discrete lookup-based embeddings. In many cases MiLES even exceeds the performance of replay techniques despite that fact that MiLES adds minimal computational overhead, or reduces overhead,

whereas the latter require more computation. It can also be seen that MiLES can be paired with replay techniques to combine their performance benefits. For the more sophisticated TUL models (MainTUL, T3S, TULHOR and TULVAE), a combination of FIFO and MiLES yields the best performance across all metrics on the Foursquare datasets.

Table 21: Top-1 accuracy, top-5 accuracy and macro F1 score for embedding and general online learning techniques for all TUL approaches.

| Dataset | Foursquare-NYC | | | Foursquare-TKY | | | GeoLife | | |
|---|---|---|---|---|---|---|---|---|---|
| Method | Acc@1 | Acc@5 | Macro F1 | Acc@1 | Acc@5 | Macro F1 | Acc@1 | Acc@5 | Macro F1 |
| BiTULER | | | | | | | | | |
| Lookup | 60.12 | 67.20 | 57.83 | 63.16 | 74.91 | 61.06 | 37.56 | 70.85 | 26.69 |
| Linear | 55.75 | 64.45 | 53.08 | 60.63 | 73.45 | 58.20 | 37.28 | 71.13 | 23.33 |
| Fourier | 60.21 | 70.38 | 58.07 | 61.84 | 75.91 | 59.38 | **57.52** | **84.66** | **38.55** |
| MiLES | **61.61** | **70.78** | 59.54 | **64.56** | **77.76** | **62.37** | 46.28 | 77.83 | 32.96 |
| Balanced | 60.04 | 66.95 | 57.95 | 63.08 | 74.56 | 60.89 | 36.91 | 67.83 | 25.28 |
| FIFO | 59.89 | 66.93 | 57.93 | 62.99 | 74.55 | 60.87 | 36.74 | 68.50 | 25.73 |
| FIFO + MiLES | **61.61** | 70.68 | **59.76** | 64.39 | 77.38 | 62.17 | 45.62 | 76.32 | 32.05 |
| DeepTUL | | | | | | | | | |
| Lookup | 58.72 | 65.48 | 56.60 | 61.17 | 72.49 | 59.10 | 36.32 | 72.64 | 29.82 |
| Linear | 55.86 | 64.21 | 53.58 | 59.75 | 72.12 | 57.45 | 35.74 | 72.44 | 26.61 |
| Fourier | 58.35 | 67.95 | 56.25 | 59.31 | 73.24 | 56.93 | **55.27** | **86.27** | **41.14** |
| MiLES | 59.77 | 68.52 | 57.86 | 62.01 | **74.86** | 59.87 | 44.92 | 78.95 | 35.79 |
| Balanced | 59.09 | 65.82 | 57.14 | 61.34 | 72.49 | 59.18 | 34.60 | 65.42 | 26.94 |
| FIFO | 58.86 | 65.65 | 56.91 | 61.20 | 72.47 | 59.09 | 37.89 | 71.62 | 29.75 |
| FIFO + MiLES | **60.00** | **68.70** | **58.16** | **62.21** | 74.77 | **60.02** | 47.01 | 78.53 | 36.09 |
| MainTUL | | | | | | | | | |
| Lookup | 55.67 | 62.61 | 53.01 | 59.53 | 71.89 | 57.09 | 34.00 | 70.26 | 21.76 |
| Linear | 51.48 | 60.34 | 48.19 | 57.59 | 71.04 | 54.67 | 33.48 | 69.98 | 20.25 |
| Fourier | 56.20 | 66.85 | 53.49 | 57.71 | 72.86 | 54.68 | **51.83** | **82.46** | **32.39** |
| MiLES | 57.11 | 66.94 | 54.54 | 61.15 | 75.34 | 58.54 | 42.19 | 76.44 | 27.27 |
| Balanced | 56.34 | 63.55 | 54.26 | 59.61 | 72.02 | 57.06 | 33.01 | 64.27 | 20.32 |
| FIFO | 56.11 | 63.38 | 54.14 | 59.63 | 72.15 | 57.47 | 35.81 | 69.90 | 22.08 |
| FIFO + MiLES | **57.71** | **67.65** | **55.64** | **61.30** | **75.51** | **58.92** | 44.59 | 76.87 | 28.60 |
| T3S | | | | | | | | | |
| Lookup | 52.98 | 60.28 | 49.50 | 56.41 | 69.26 | 53.24 | 35.25 | 71.11 | 21.52 |
| Linear | 39.56 | 47.52 | 35.49 | 38.65 | 52.21 | 34.85 | 33.53 | 69.51 | 19.97 |
| Fourier | 52.21 | 62.61 | 49.01 | 53.67 | 68.72 | 50.39 | **51.35** | **82.74** | **32.29** |
| MiLES | 54.69 | 63.42 | 51.60 | 58.10 | 72.05 | 55.16 | 42.23 | 75.90 | 26.70 |
| Balanced | 54.14 | 61.37 | 51.02 | 56.74 | 69.25 | 53.64 | 39.04 | 72.90 | 25.87 |
| FIFO | 54.71 | 62.06 | 52.26 | 57.43 | 69.85 | 54.94 | 37.48 | 71.68 | 23.38 |
| FIFO + MiLES | **56.10** | **64.67** | **53.77** | **59.05** | **72.30** | **56.54** | 45.12 | 77.23 | 29.01 |
| TULHOR | | | | | | | | | |
| Lookup | 53.85 | 61.13 | 50.40 | 56.61 | 69.61 | 53.45 | 34.65 | 72.46 | 24.92 |
| Linear | 46.90 | 55.26 | 42.85 | 51.77 | 65.17 | 48.29 | 32.53 | 70.22 | 22.22 |
| Fourier | 53.34 | 63.71 | 50.17 | 54.70 | 69.71 | 51.39 | **55.78** | **86.46** | **40.39** |
| MiLES | 55.56 | 64.29 | 52.51 | 58.46 | 72.41 | 55.48 | 42.85 | 77.49 | 30.61 |
| Balanced | 54.41 | 61.71 | 51.15 | 56.98 | 69.58 | 53.77 | 38.36 | 73.92 | 28.41 |
| FIFO | 55.02 | 62.40 | 52.48 | 57.75 | 70.24 | 55.13 | 36.90 | 72.33 | 26.82 |
| FIFO + MiLES | **56.44** | **65.00** | **54.01** | **59.32** | **72.64** | **56.75** | 44.65 | 77.77 | 32.21 |
| TULVAE | | | | | | | | | |
| Lookup | 59.79 | 66.77 | 57.32 | 55.82 | 66.23 | 51.68 | 37.08 | 70.45 | 25.25 |
| Linear | 54.90 | 63.46 | 51.88 | 52.48 | 63.52 | 47.53 | 37.57 | 70.43 | 23.56 |
| Fourier | 59.41 | 69.29 | 56.93 | 55.31 | 68.23 | 50.78 | **55.76** | **83.36** | **35.45** |
| MiLES | 61.57 | 70.50 | 59.36 | 58.89 | 70.66 | 55.01 | 45.10 | 77.02 | 29.99 |
| Balanced | 59.09 | 66.13 | 56.79 | 54.73 | 64.95 | 50.30 | 36.49 | 65.89 | 25.33 |
| FIFO | 59.69 | 66.79 | 57.72 | 57.71 | 68.31 | 54.35 | 39.14 | 71.21 | 27.18 |
| FIFO + MiLES | **61.78** | **70.60** | **59.86** | **60.75** | **72.56** | **57.67** | 48.17 | 78.25 | 33.29 |

# B   In-Depth Analysis of MiLES

## B.1   Ablation of Individual Embedding Levels

To isolate the contribution of different levels of spatial granularity, we evaluated BiTULER models trained with single embedding levels from MiLES. Figure 9 shows the top-5 accuracy over the first 10,000 trajectories.

Initially, the models using coarser, shared embeddings (Levels 2 & 3) adapt more quickly, outperforming the POI-only model (Level 0). After approximately 1,500 trajectories, the higher-specificity Level 0 model achieves better peak accuracy. However, it also exhibits greater sensitivity to distribution shifts (e.g., dips around 4k and 6k samples). The full MiLES model successfully combines the stability of the high-level embeddings with the peak performance of the fine-grained ones.

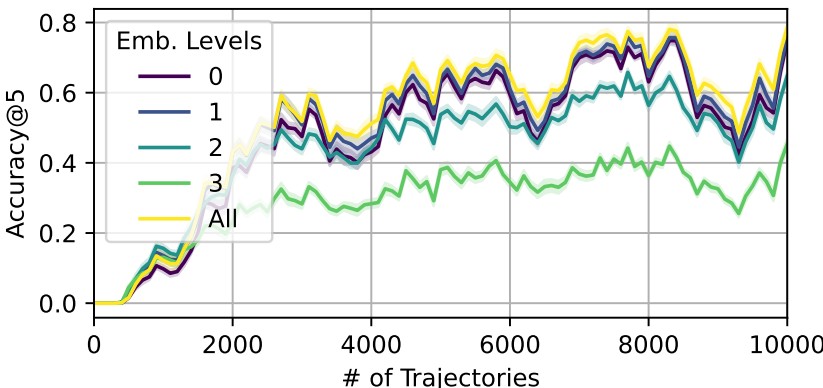

Figure 9: Top-5 accuracy of BiTULER models using individual MiLES embedding levels on Foursquare-NYC.

## B.2   Ablation of Grid Geometry

To validate our choice of a hexagonal grid over a more conventional square grid, we re-ran our evaluations of MiLES using a square grid tiling while keeping all other hyperparameters identical to the main experiments.

As shown in Table 22, the hexagonal grid model consistently outperformed its square-grid counterpart. The performance gap was particularly pronounced on the GeoLife dataset, which does not feature POI information and therefore relies solely on grid-based embeddings. While specifically tuning the hyperparameters of MiLES for a square grid would likely yield improvements, these results empirically support the theoretical advantages of hexagonal grids for better representing spatial proximity in mobility tasks (Ke et al., 2019).

Table 22: Performance comparison of MiLES with hexagonal vs. square grids, using the BiTULER backbone. Results show top-1 accuracy, top-5 accuracy and macro F1. Best results in bold.

| Dataset | Grid Shape | Acc@1 | Acc@5 | F1 |
|---|---|---|---|---|
| Foursquare-NYC | Hexagon | **61.61** | **70.78** | **59.54** |
| | Square | 61.50 | 70.70 | 59.41 |
| Foursquare-TKY | Hexagon | **62.72** | **75.95** | **60.40** |
| | Square | 62.58 | 75.92 | 60.26 |
| GeoLife | Hexagon | **46.28** | **77.83** | **32.96** |
| | Square | 37.01 | 72.82 | 25.39 |

## B.3 Embedding Design vs. Grid Configuration on GeoLife

In Table 6, the DeeprETA-style embedding (Hu et al., 2022) is slightly ahead of MiLES on GeoLife. As noted there, each method's grid configuration (grid shape, number of levels, and cell growth factor) is tuned once on Foursquare-TKY and transferred unchanged to every dataset, mirroring the realistic online setting in which no in-distribution data is available for tuning ahead of the stream. Under this protocol, the configuration selected for DeeprETA (a square grid with three levels and a growth factor of four) happens to suit GeoLife's larger spatial extent better than the one selected for MiLES, and Table 22 already shows that the grid configuration alone strongly affects MiLES on GeoLife. To separate the effect of the grid from that of the embedding design, we re-ran MiLES's embedding (variable per-level dimensions and learnable level weights) at DeeprETA's own tuned grid, so the two methods differ only in the embedding.

As Table 23 shows, at this matched grid MiLES's embedding design outperforms the DeeprETA-style embedding on GeoLife across all six backbones, recovering an average of 0.84 pp. of top-1 accuracy and surpassing MiLES's own transferred-grid result (43.93). This confirms that MiLES's variable dimensions and learnable weighting remain advantageous even on POI-free GeoLife, consistent with the -WL and -VD ablations (Figure 4), and that the small aggregate gap in Table 6 reflects grid transfer rather than embedding design.

Table 23: Controlled comparison on GeoLife with the grid configuration held fixed (square grid, three levels, growth factor four, tuned for the DeeprETA-style embedding). MiLES retains its variable per-level dimensions and learnable level weights. The DeeprETA-style embedding uses uniform per-level dimensions without learnable weights. Top-1 accuracy, top-5 accuracy and macro F1 [%], averaged over five seeds. Best per backbone in bold.

| Backbone | DeeprETA-style | | | MiLES | | |
|---|---|---|---|---|---|---|
| | Acc@1 | Acc@5 | F1 | Acc@1 | Acc@5 | F1 |
| BiTULER | 46.92 | 79.29 | 32.96 | **47.64** | **79.42** | **33.69** |
| TULVAE | 44.04 | 76.35 | 27.92 | **44.87** | **76.87** | **28.13** |
| DeepTUL | 45.02 | 80.04 | 35.86 | **46.50** | **80.96** | **36.61** |
| MainTUL | 43.30 | 78.18 | 27.99 | **44.24** | **78.58** | **28.10** |
| T3S | 43.40 | 77.75 | 27.63 | **44.28** | **78.39** | **28.37** |
| TULHOR | 43.86 | 79.35 | 31.42 | **44.08** | **79.47** | **31.70** |
| Average | 44.43 | 78.49 | 30.63 | **45.27** | **78.95** | **31.10** |

## B.4 Aggregation of Embedding Levels

To validate our choice of concatenation for aggregating MiLES's embedding levels, we compared it against summation. Figure 10 shows the results on the 399-user Foursquare-NYC dataset using BiTULER with fixed level weights.

When summing embeddings, adding more levels degrades top-2 accuracy, suggesting that the less-precise, high-level embeddings interfere with the fine-grained POI embeddings crucial for distinguishing individual users. While summation improves top-5 accuracy in some cases (as coarser features can help identify user groups), the concatenation-based approach consistently achieves the highest performance for both metrics. This empirically supports our design choice for MiLES.

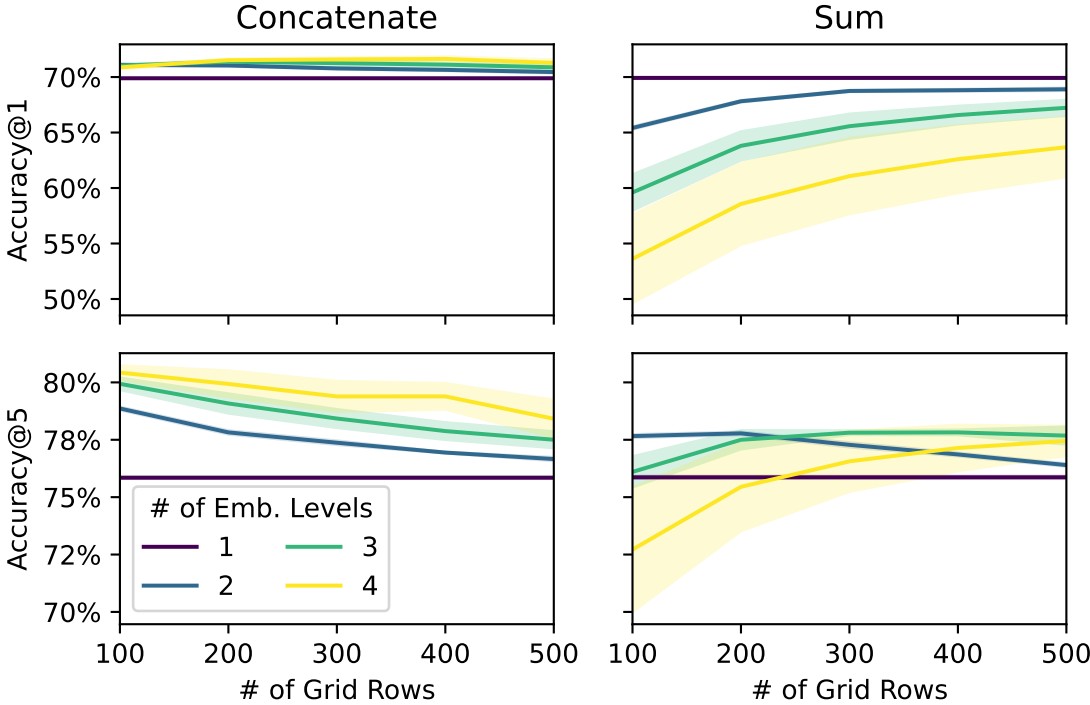

Figure 10: Performance on the 399 user Foursquare-NYC variant, depending on the grid resolution, the number of embedding levels and whether level-specific embeddings were concatenated or summed. BiTULER was used as the classifier and level weights $w_l$ were fixed. Shaded areas represent the $1\sigma$ range.

### B.5 Components of the Per-Level Contribution

Figure 7 tracks the per-level contribution share. Figure 11 decomposes the underlying magnitude $g_l = w_l \|\boldsymbol{Z}_l\| \|\boldsymbol{W}_l\|$ into its three factors for MiLES and $-$WL on the 800 user Foursquare-NYC dataset with BiTULER. The embedding norms $\|\boldsymbol{Z}_l\|$ and the input-weight norms $\|\boldsymbol{W}_l\|$ are nearly identical between the two models and vary little across levels. The per-level re-weighting therefore comes almost entirely from $w_l$, which $-$WL lacks (its weights are fixed at one).

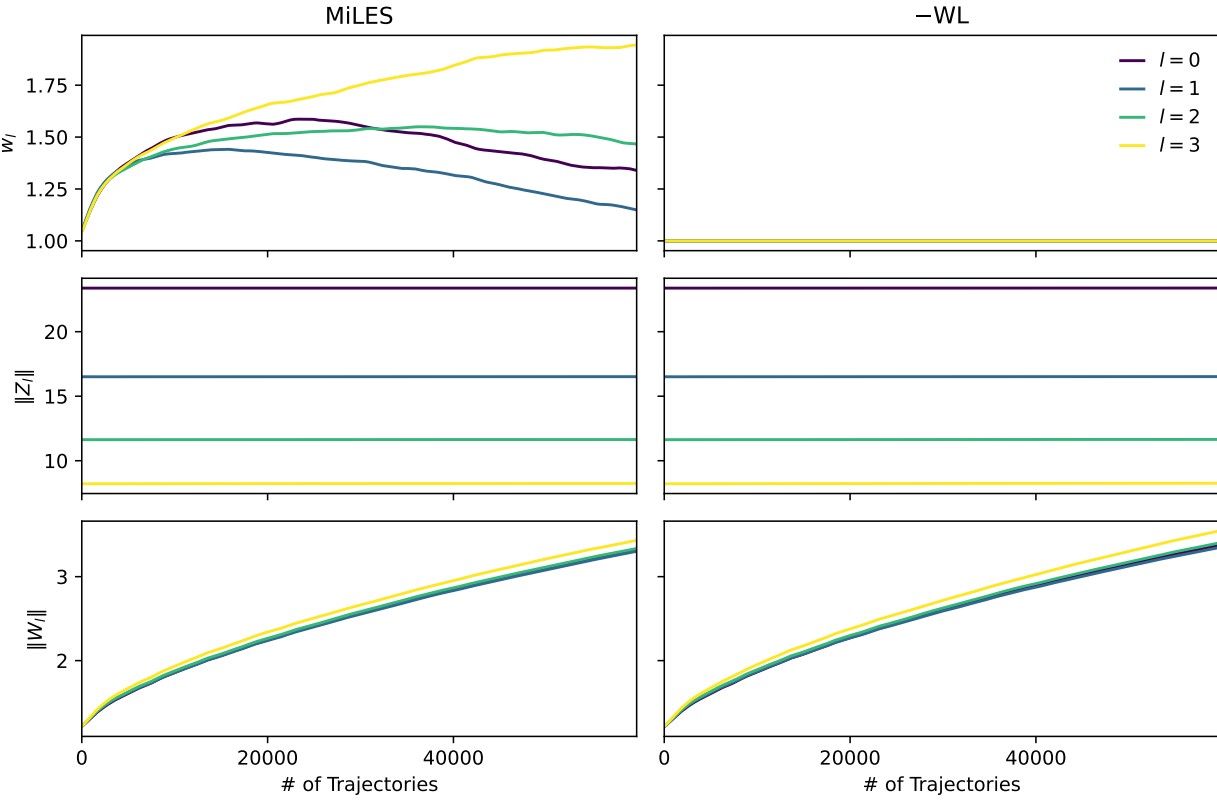

Figure 11: Absolute per-level components of the contribution magnitude $g_l = w_l \|\boldsymbol{Z}_l\|\|\boldsymbol{W}_l\|$ over the online stream, for MiLES (left) and $-$WL (right) on Foursquare-NYC (800 users, BiTULER, mean over five seeds). Rows: the level weight $w_l$, the embedding norm $\|\boldsymbol{Z}_l\|$, and the input-weight norm $\|\boldsymbol{W}_l\|$. Colors denote the level $l$. The embedding and input-weight norms barely differ between the two models or across levels, so the per-level re-weighting in Figure 7 is carried by $w_l$ alone. $-$WL keeps $w_l = 1$.

## B.6 Level Weights vs. an Effective Learning Rate

Section 4 identifies the level weights $w_l$ as a per-level forward scale: a well-conditioned single parameter controlling the magnitude at which each level enters the backbone (Figure 7). A natural alternative is that $w_l$ instead acts as an effective per-level *learning rate*, since the gradient flowing into level $l$'s embedding table is also scaled by $w_l$, speeding up or slowing down how fast that table is updated rather than how strongly it is read. We test this reading directly. Starting from the $-$WL model, we assign each level's embedding table a per-step learning rate proportional to the recorded full-MiLES trajectory $w_l(t)$, replaying the schedule the weights actually follow but as a learning-rate multiplier rather than a forward scale. As shown in Table 24, this leaves performance at the $-$WL level, recovering essentially none of the gap, whereas the forward-scale mechanism of Figure 7 accounts for it. The level weights therefore appear to act through the forward scale at which each level enters the model, not through the rate at which its embedding table is learned.

Table 24: Testing whether the level weights act as an effective per-level learning rate, on Foursquare-NYC (800 users, BiTULER, mean $\pm$ std over five seeds). Replaying each seed's recorded $w_l(t)$ as a per-step learning rate on the embedding tables (rather than as a forward scale) leaves performance at the $-$WL level, recovering essentially none of the full-MiLES gap.

|  | Acc@1 | Acc@5 | F1 |
|---|---|---|---|
| Full MiLES (learnable $w_l$) | $61.61 \pm 0.04$ | $70.78 \pm 0.04$ | $59.54 \pm 0.08$ |
| $-$WL (no level weights) | $61.00 \pm 0.09$ | $70.29 \pm 0.06$ | $58.85 \pm 0.13$ |
| $-$WL + per-step LR $\propto w_l(t)$ | $61.09 \pm 0.06$ | $70.34 \pm 0.07$ | $58.93 \pm 0.08$ |

## C  Concept Drift in Trajectory Data

As mentioned above, real-world applications based on mobility data are likely to face changes in the data distribution in the form of concept drift. Such drifts can also be found in the datasets used in this work, and we provide three complementary pieces of descriptive evidence below.

An example for concept drift in trajectory data is the emergence of new POIs. As can be seen in Figure 12, the percentage of visited unique POIs increases quickly at the start of both Foursquare datasets but continues to rise at an almost linear rate throughout both datasets. If a batch learning model were to be trained on the first half of the datasets, it would therefore have encountered only approximately 60% of unique POIs, causing it to struggle interpreting trajectories in the latter half of the datasets, which highlights the need of online learning approaches for mobility mining tasks.

Beyond the emergence of new locations, the distribution over existing POIs also shifts over time. Figure 13 shows the Kullback-Leibler divergence between POI visitation distributions in consecutive trajectory windows on Foursquare-NYC. The divergence increases steadily over the course of the stream, indicating that visitation patterns continue to change even after the initial learning phase. The peak around trajectory 40,000 coincides with a dip in rolling accuracy observed in Figure 2, suggesting that this distributional shift has a tangible impact on model performance.

Figure 14 shows that the composition of active users in GeoLife varies considerably over the data stream. The entropy of user labels within a sliding window fluctuates around a value of 3, until declining sharply after 20000 trajectories have been observed. Such variation changes the effective difficulty of the classification task over time and represents another form of non-stationarity that online models must contend with.

Figure 15 provides a visual example of concept drift found in the GeoLife dataset. While the trajectories recorded earlier in the collection period and shown in a darker hue are relatively evenly distributed across the area, the later trajectories are much more concentrated around the major transportation axes. This effect could be caused by a change in either user behavior or data collection methodology.

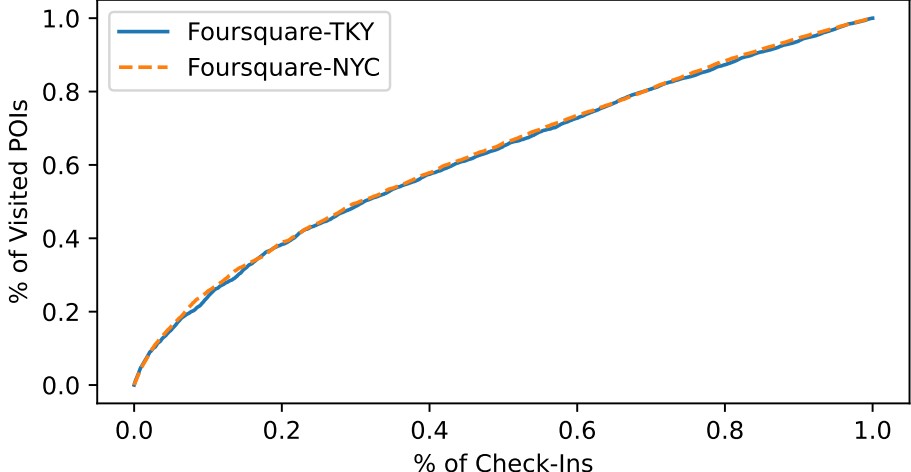

Figure 12: Growth of the observed POI vocabulary relative to the percentage of processed check-ins. The lack of saturation indicates that the model continuously encounters new, unlearned locations throughout the stream.

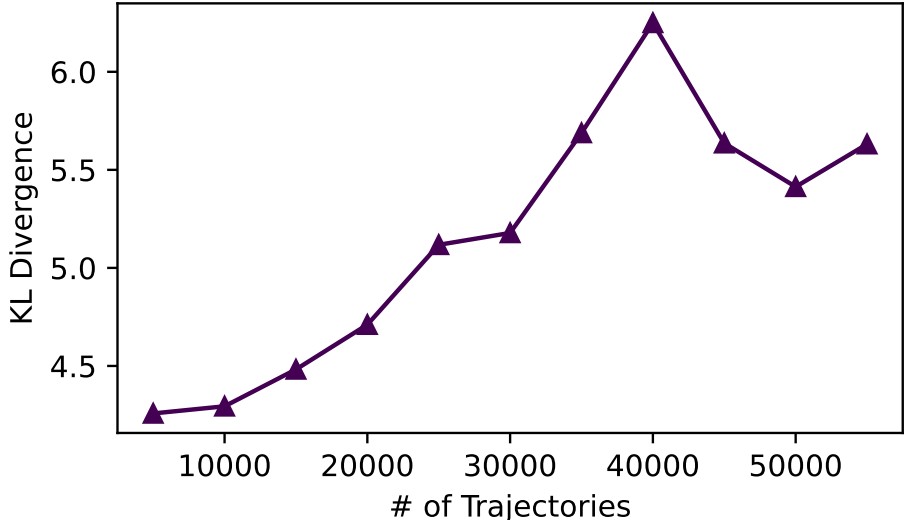

Figure 13: KL divergence between POI visitation distributions in consecutive non-overlapping windows of 5,000 trajectories on Foursquare-NYC. Higher values indicate greater distributional shift between adjacent segments of the stream.

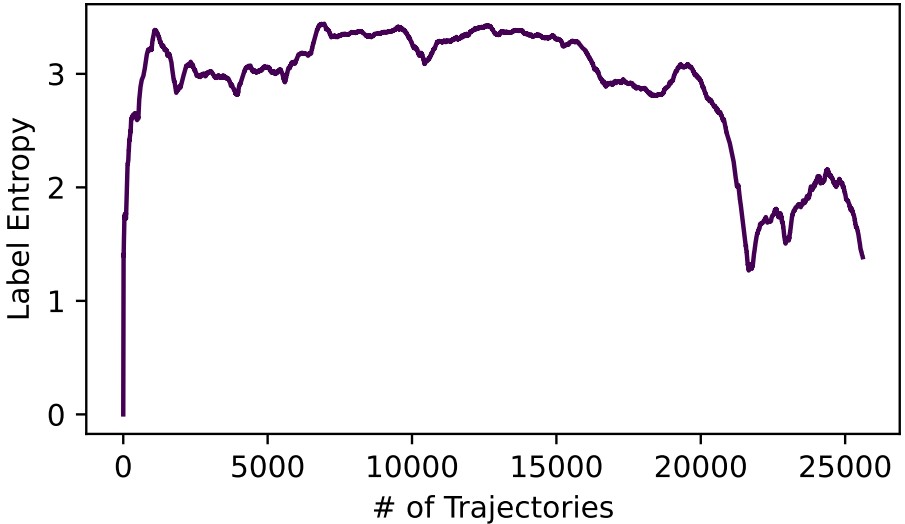

Figure 14: Shannon entropy of user labels within a sliding window of 1,000 trajectories on GeoLife. Fluctuations reflect changes in the number and concentration of active users over the data stream.

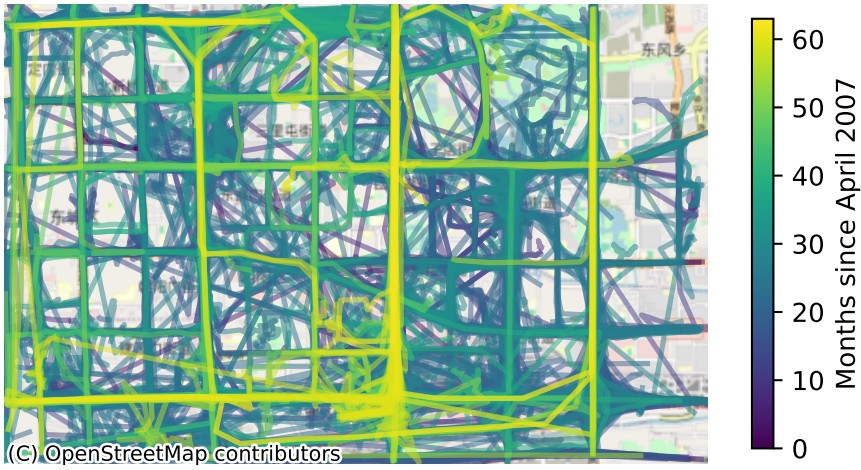

Figure 15: Distribution of trajectories in the GeoLife dataset over time for highly frequented area. Trajectories are color-coded according based on their time of occurrence.

