# OpenReview forum: "Multi-Level Spatial Embedding Sharing for Enhanced Online Trajectory-User Linking"
_TMLR — Under review for TMLR_

### Review · Reviewer_EQjN · 2026-06-12

**Summary Of Contributions:**

This paper presents the first systematic study of online learning for Trajectory-User Linking (TUL), a setting where trajectory data arrives sequentially and must be processed in real time without retaining historical data. To address the challenge of gradient sparsity caused by the large number of unique Points of Interest (POIs) and the single-pass nature of online learning, the authors propose Multi-Level Spatial Embedding Sharing (MiLES). MiLES combines POI-level embeddings with multiple coarse-to-fine hexagonal grid-based embeddings, using a learnable weighting mechanism to balance specificity and gradient density. The method significantly reduces embedding parameters (by up to 38%) while improving top-1 accuracy by up to 24% across various TUL models. The paper further validates MiLES on destination prediction, demonstrating its generality beyond TUL. Extensive experiments on three real-world datasets, including concept drift analysis and ablation studies, confirm that online learning outperforms periodic batch retraining, and that MiLES consistently improves model robustness and efficiency.

**Additional Comments:**

See Requested Changes

**Audience:**

Yes

**Audience Explanation:**

See Summary Of Contributions

**Broader Impact Concerns:**

Based on the provided manuscript, the paper does not contain a dedicated "Broader Impact Statement" section. Given the nature of the research, this is a significant omission that should be addressed.

Concerns regarding ethical implications requiring the addition of a Broader Impact Statement:

The paper focuses on Trajectory-User Linking (TUL) in an online learning setting, which inherently deals with highly sensitive personal mobility data. While the authors briefly mention privacy benefits (e.g., discarding data immediately after updates), several critical ethical and societal implications are not sufficiently discussed.

**Claims And Evidence:**

Yes

**Claims Explanation:**

See Summary Of Contributions

**Requested Changes:**

This manuscript presents a timely and systematic study of online learning for Trajectory-User Linking (TUL), a problem setting that has been largely overlooked in existing literature. The proposed Multi-Level Spatial Embedding Sharing (MiLES) method effectively addresses the gradient sparsity issue in online TUL by combining POI-level embeddings with multi-scale grid-based embeddings and learnable weighting. The authors conduct extensive experiments on multiple real-world datasets, evaluate a wide range of TUL backbones, and provide compelling evidence of performance improvements, particularly in online learning scenarios. The inclusion of a formal analysis on the density-information trade-off and concept drift analysis further strengthens the empirical foundation of the work. However, several major concerns need to be addressed before the manuscript can be considered for publication.

Major concerns:

1. While the manuscript compares MiLES with FIFO replay, random replay, and basic adaptive optimizers (DoG, AdamHD), it lacks comparisons with more advanced online learning techniques specifically designed for sequence modeling or trajectory data. For example, methods such as Streaming LDA, Online HMMs, or recent streaming graph-based clustering approaches have not been evaluated. Furthermore, although the paper briefly mentions Space2Vec and Fourier features, it does not provide direct experimental comparisons against these multi-scale spatial encoding methods within the same framework. Given that MiLES is fundamentally a spatial embedding technique, comparing it against a properly tuned Space2Vec baseline would significantly strengthen the claimed contributions.

2. The formal analysis in Appendix B.1 (Equations 5-7) provides a rigorous derivation of the entropy loss due to embedding sharing. However, this analysis is relegated to the appendix and is not discussed in the main body of the paper. Integrating this theoretical insight into Section 4 or Section 6 would help readers understand the fundamental trade-off MiLES balances and why the multi-level design is necessary. Additionally, the analysis assumes uniform probabilities p(z(k)), which may not hold in real-world long-tail distributions. The authors should discuss how this assumption affects the validity of the information loss quantification.

3. In Table 10, MiLES achieves a mean error reduction of 0.13 km on Foursquare-NYC for CNN, but on Foursquare-TKY, the mean error increases by 0.05 km compared to the default embedding. The authors attribute this to "local connectivity of convolutional kernels already causing some similarity between embeddings of nearby POIs." While this explanation is plausible, it raises concerns about the generalizability of MiLES across different network architectures. A more detailed analysis is needed to understand why CNNs are less responsive to MiLES, and whether this limitation can be mitigated through architectural modifications or hyperparameter tuning.

4. The authors state that hyperparameters were tuned using the first 5,000 trajectories of the 400-user Foursquare-TKY dataset and then applied to the remaining stream. While this is a common practice in online learning evaluations, it raises concerns about temporal data leakage. If the hyperparameters are optimized for the specific distribution of the first 5,000 trajectories, they may overfit to that initial segment, potentially biasing performance on the later segments that exhibit different distributional characteristics (as evidenced by the concept drift analysis in Appendix C). The authors should consider adopting a rolling time-series cross-validation approach for hyperparameter selection or provide a sensitivity analysis showing that performance is robust to different tuning windows.

5. The manuscript fixes the total embedding dimension across all methods to ensure a fair comparison. However, when MiLES introduces multiple levels (e.g., four levels), the actual per-level embedding dimensions are reduced to keep the total constant. This introduces a confounding factor: the performance gain of MiLES could partially stem from the beneficial effect of having multi-level features, but also from the forced reduction in POI embedding dimensionality. The ablation study in Figure 4 removes the variable dimensionality (-VD) and shows a large drop in macro F1, suggesting that dimension allocation is critical. However, a more systematic ablation varying the total embedding dimension or the dimension allocation ratio across levels would help disentangle these effects.

Minor concerns:

1. In Section 3.4, the reference to "ASAM" (Adaptive Sharpness-Aware Minimization) is not fully explained. The manuscript introduces Delayed Sharpness Optimization (DSO) but does not provide the underlying mathematical formulation of ASAM within the main text. Readers unfamiliar with ASAM would struggle to understand how DSO extends it. Adding the ASAM formulation or a brief summary would improve accessibility.

2. The definition of "recovery" in Table 4 is somewhat ambiguous. The text states that recovery denotes the number of post-drift trajectories until rolling accuracy returns to 95% of the pre-drift value. However, the pre-drift value is not explicitly reported, and the 95% threshold seems arbitrary. The authors should justify this threshold or provide alternative metrics, such as the area under the recovery curve.

3. Several figures (e.g., Figure 4, Figure 6) are referenced in the text but could benefit from more detailed captions that clearly state what each subplot represents. For instance, in Figure 4, the x-axis labels are ambiguous in terms of whether they represent number of embedding levels or grid resolution.

4. The paper contains minor grammatical issues (e.g., "the models ability" in Section 6.1) and inconsistent capitalization (e.g., "Grid" vs. "grid" in Figure 5 caption). A careful proofreading is recommended.

5. In Table 5, the wall time for GeoLife is higher for MiLES (11.66 ms) than for the default embedding (11.27 ms), despite the lower parameter count. The authors attribute this to "optimized implementation." A more detailed explanation of the computational overhead and why it increases despite fewer parameters would be helpful for practitioners.

6. The literature review should be expanded to include recent advances in trajectory modeling, social group detection, and multimodal feature fusion that are closely related to the proposed MiLES framework for online Trajectory-User Linking. For example, such as [a] Visual Exposes You: Pedestrian Trajectory Prediction Meets Visual Intention, which integrates visual intention for enhanced trajectory prediction; [b] Multi-Level Drowsiness Detection Based on Deep Feature Fusion of Eye and Head Pose, which leverages deep feature fusion for multi-level state analysis; and [c] SegTraj: Segmented-Trajectory-Aware Spatio-Temporal GCN for Social Group Detection, which employs segmented-trajectory-aware spatio-temporal graph networks for modeling group mobility. These works encompass trajectory modeling, multimodal feature fusion, and spatio-temporal graph reasoning, all of which are closely related to the representation learning, embedding aggregation, and spatial-temporal modeling challenges addressed by MiLES.

---

> ### Author Response · Authors · 2026-07-05
>
> We thank the reviewer for the detailed and constructive review. As the comments are extensive, we split our response across three comments.
>
> **Comparison with advanced online learning techniques (Streaming LDA, Online HMMs, streaming graph clustering)**
>
> We appreciate these concrete suggestions. MiLES is a model-agnostic embedding component rather than a standalone online learner. The space of online-learning methods is effectively open-ended, so rather than aim for exhaustive coverage we evaluate several prevalent representatives from different families that are applicable to deep TUL backbones, namely FIFO and random experience replay together with the adaptive optimizers DoG and AdamHD (Table 6). We further show that MiLES combines effectively with each of them, so it is best regarded as orthogonal and complementary to this line of work rather than competing with it. The suggested methods sit outside this class, having been introduced for different tasks (for example Streaming-LDA for online topic modeling and streaming graph clustering for unsupervised clustering), and do not provide the per-model embedding component that MiLES does for the six TUL architectures we evaluate (Table 2).
>
> **Space2Vec / Fourier baseline comparison**
>
> Our evaluation already includes a Fourier-feature baseline (the Fourier row in Table 6, with per-dataset results in Table 21), evaluated under the same online protocol and backbones as every other method. This encoding is the grid variant of Space2Vec and differs from it only in the wavelength range, which we tune. While revising this comparison we found that the Fourier rows had used an untuned frequency scale rather than the tuned value from our hyperparameter table, and we have re-run them at the tuned scale across all six backbones and three datasets. We additionally add the theory variant of Space2Vec, which uses a multi-directional sinusoidal basis, as an additional baseline (also Table 6).
>
> With the corrected scale both continuous encoders are strong baselines and stay within roughly 1.5 pp. of each other on every dataset. On the POI-rich Foursquare datasets MiLES still leads them across all backbones and metrics. On the POI-free GeoLife dataset, where the grid cannot resolve nearby points, the continuous encoders outperform MiLES. We have removed the claim that MiLES outperforms all embeddings on all datasets and revised the discussion.
>
> Summary of changes:
>
> - Re-ran the Fourier baseline (the grid variant of Space2Vec) at the tuned frequency scale across all backbones and datasets, updating Tables 6 and 21.
> - Added the Space2Vec theory variant as an explicit baseline (Table 6) and noted in the caption and related work that the Fourier baseline is its grid variant.
> - MiLES leads both continuous encoders on the Foursquare datasets across all backbones, while on GeoLife they lead MiLES. Removed the claim that MiLES outperforms all embeddings on all datasets, and reframed the contributions and conclusion to present MiLES as an effective alternative to the standard POI-only lookup embedding, with a continuous encoder more effective on denser GPS data such as GeoLife.
>
> **Move formal analysis to main text and discuss the distributional assumption**
>
> We agree and have moved the formal analysis (formerly Appendix B.1) into the approach section. The derivation does not assume a uniform distribution, as it holds for arbitrary $p(z(k))$. To make the role of the distribution explicit, we added a factorization showing that the information loss equals the cell mass times the entropy of the within-cell distribution, which is maximized in the uniform case and smaller under the long-tailed distributions typical of mobility data.
>
> Summary of changes:
>
> - Moved the formal density-information analysis into the approach section and added a discussion of how the loss depends on the visitation distribution (no uniformity assumed, uniform is the worst case, and long-tailed visits reduce the loss).

---

> > ### Author Response · Authors · 2026-07-05
> >
> > **CNN destination prediction on Foursquare-TKY**
> >
> > We thank the reviewer for flagging this case, which prompted us to examine it more closely. To put its magnitude in perspective, across the 30 model $\times$ dataset $\times$ task scenarios we evaluate, the CNN on Foursquare-TKY is the single case where MiLES does not improve over the default embedding, and the drop is 0.05 km, under $1\%$ of the 6.57 km baseline error. MiLES improves the remaining 29.
> >
> > While we do not think this points to a broader generalization issue, the reviewer is right that it deserves an explanation, which we now provide in the results. At a fixed total embedding dimension, MiLES reallocates capacity from the POI level to coarser shared levels, which the new dimension budget sweep (the reviewer's fifth point) shows is harmful in isolation. The coarse levels normally more than compensate by reducing gradient sparsity. For the CNN, however, whose kernels already smooth nearby POIs through their receptive field, this spatial sharing appears to be partially redundant, so the lost POI-level capacity is no longer offset.
> >
> > Even here MiLES remains attractive, trailing by less than 1% of the baseline error while retaining its parameter-efficiency advantage, so the trade-off stays favorable rather than becoming a regression. The gains are also conservative, since MiLES uses one shared configuration tuned for the BiLSTM model rather than per backbone, so a CNN-specific configuration could plausibly close the gap. We nonetheless state the scope of our evidence plainly in the limitations, noting that MiLES helps across the recurrent, attention-based and convolutional backbones we test but cannot be claimed to benefit every architecture family.
> >
> > Summary of changes:
> >
> > - Added an explanation of the TKY-CNN result to the destination-prediction results, and a measured limitations sentence acknowledging we cannot claim MiLES benefits every architecture family.
> >
> > **Hyperparameter tuning and temporal leakage**
> >
> > The protocol does not introduce temporal leakage in its conventional sense. Hyperparameters are selected on the earliest 5,000 trajectories of the stream and then frozen, so information flows strictly forward in time and no future or held-out data informs the tuning. Hyperparameter selection is also not the focus of this work. We adopt a common practice and apply it uniformly to all models and methods, so any disadvantage from transferring a configuration across datasets affects the baselines equally and does not favor MiLES.
> >
> > To address the overfitting concern directly, we re-ran the MiLES sweep on a different 5,000-trajectory window from the middle of the Foursquare-TKY stream (trajectories 20,000 to 25,000). The resulting configuration differs in three of its four hyperparameters, yet when both are trained from scratch over the full stream and compared on a common held-out tail (index 25,000 onward, unseen by either tuning run), they stay within one percentage point on every metric, with the re-tuned configuration slightly ahead (top-1 73.91 vs. 74.25, top-5 85.92 vs. 86.66, macro F1 69.44 vs. 69.93). Performance is therefore likely insensitive to the tuning window, and tuning on the earliest segment appears to be the conservative choice.
> >
> > Summary of changes:
> >
> > - Clarified in the protocol that hyperparameters are selected only on the earliest segment and frozen thereafter, using no future or held-out information.
> > - Added a tuning-window sensitivity experiment in the appendix (\Cref{tab:window_robustness}).
> >
> > **Disentangling multi-level structure from dimension reduction**
> >
> > The reviewer raises a legitimate concern. Because MiLES allocates only a fraction of its total embedding dimension to the POI level, its gains could in principle reflect implicit regularization from a smaller POI embedding rather than the added levels. We test this with a total-dimension sweep (Table 8) from 128 to 2048 with the hidden size fixed, comparing MiLES against the single-level POI lookup and the Fourier encoder on Foursquare-NYC (800 users). Shrinking the single-level POI lookup monotonically degrades performance (top-1 falls from 60.11% at $d=1024$ to 44.93% at $d=128$). MiLES allocates only 547 of its 1024 dimensions to the POI level yet reaches 61.60% top-1, surpassing the single-level baseline that gives all 1024 dimensions to the POI embedding, and it leads at every budget and metric. This rules out implicit regularization, since the coarser levels more than compensate for the smaller POI-embeddings, so MiLES's gains come from the added spatial information.
> >
> > Summary of changes:
> >
> > - Added a total-dimension budget sweep (Table 8) comparing MiLES, the single-level POI lookup, and the Fourier encoder from 128 to 2048 dimensions, ruling out implicit regularization, since shrinking the POI lookup degrades performance while MiLES leads at every budget.

---

> > > ### Author Response · Authors · 2026-07-05
> > >
> > > **ASAM / Delayed Sharpness Optimization**
> > >
> > > We could not locate any reference to ASAM, Delayed Sharpness Optimization, or a Section 3.4 in our manuscript, which does not discuss sharpness-aware optimization in any form. We suspect this comment may have been intended for a different submission, and would be happy to address it if the reviewer can point us to the relevant part of our paper.
> > >
> > > **Recovery definition in Table 4**
> > >
> > > We have expanded Table 4 to make the recovery definition self-contained and switched from a 95% to a 100% recovery threshold to avoid confusion. We added the pre-drift accuracy itself (Acc@1/Acc@5 of 52.94%/60.76% for the default embedding and 55.68%/65.30% for MiLES) and the average rolling accuracy over the post-drift-to-recovery window. The latter directly addresses the reviewer's suggestion of a curve-based summary, as it is a normalized area under the recovery curve. MiLES has a higher pre-drift accuracy, a higher post-drift floor, and a higher average accuracy throughout the recovery window on both metrics, so its advantage during the drift episode is consistent.
> > >
> > > Summary of changes:
> > >
> > > - Switched the Table 4 recovery threshold from 95% to 100% of the pre-drift value and added pre-drift accuracy and average recovery-window accuracy as new columns, updating the caption and text.
> > >
> > > **Figure captions**
> > >
> > > We have expanded the captions of the three referenced figures. The ablation scatter plot now glosses each label (-L1/-L2/-L3 remove an embedding level, -VD removes the level-dependent dimensions, -WL removes the learnable level weights). The grid-resolution figure now attributes the number of grid rows to the x-axis and the embedding levels to color. The level-weight dynamics figure now attributes the level weights to the left axis and the number of unique users to the right axis.
> > >
> > > **Grammar and capitalization**
> > >
> > > We thank the reviewer for the close reading. We carried out a proofreading pass but could not find the specific "the models ability" typo described (the manuscript reads "the model's ability"). On "Grid" vs. "grid", we believe this refers to the axis label "\# of Grid Rows" in Figure 5. We use title case for all axis labels as a consistent convention while the body text uses lowercase "grid" as a common noun, so this is intentional.
> > >
> > > **Wall time explanation for GeoLife**
> > >
> > > We have added a sentence noting that GeoLife trajectories are considerably longer on average than those in the Foursquare datasets (Table 1), so MiLES's per-check-in overhead from multiple lookups and aggregation is incurred more often per trajectory. This compounds the effect already noted, namely that GeoLife's default embedding has a comparatively low parameter count, leaving little room for MiLES's efficiency gains to outweigh the overhead.
> > >
> > > **Literature suggestions**
> > >
> > > We have expanded the related-work section to situate MiLES within the broader trajectory-modeling landscape, including the suggested works [a-c].
> > >
> > > **Broader impact statement**
> > >
> > > The manuscript already contains a dedicated Broader Impact Statement (following the Limitations section), discussing the dual-use nature of improved TUL models and recommending mitigations such as differential privacy and anonymization, and noting that our online setting allows trajectory data to be discarded immediately after each update. We keep this concise as the privacy implications apply to mobility-data mining broadly rather than to MiLES specifically, and are treated in greater depth in the works we cite. To improve visibility, we have promoted it to a subsection.

---

### Review · Reviewer_TiBh · 2026-06-15

**Summary Of Contributions:**

The paper studies trajectory–user linking in an online-learning setting, where a model must predict the user associated with each incoming trajectory and subsequently update itself using the revealed label. The authors identify sparse updates to conventional POI embeddings as an important difficulty in this setting, particularly because each trajectory is observed only once and newly introduced or infrequent locations receive little training signal.

To address this problem, the paper proposes Multi-Level Spatial Embedding Sharing (MiLES), which represents each check-in using a combination of POI-specific or fine-resolution embeddings and progressively coarser spatial-grid embeddings. This provides a spectrum between highly specific but sparsely updated representations and more broadly shared representations that receive denser updates. MiLES is designed as a modular replacement for the spatial embedding component and is evaluated across several existing TUL architectures and multiple datasets. The results show generally consistent performance improvements, particularly on GeoLife, together with reductions in the number of embedding parameters.

The paper is well motivated, and the online formulation addresses a practically relevant but relatively unexplored setting. The proposed method is simple, compatible with multiple architectures, and supported by a broad experimental evaluation.

**Audience:**

Yes

**Audience Explanation:**

The paper should be relevant to researchers working on online and continual learning, trajectory and mobility modeling, spatial representation learning, recommendation, and parameter-efficient adaptation.

**Claims And Evidence:**

Yes

**Claims Explanation:**

The main claims of the paper are supported by reasonably clear and convincing evidence. MiLES is evaluated across multiple TUL architectures, datasets, and metrics under a consistent prequential online-learning protocol. The improvements over the original spatial embeddings are generally consistent, and the larger gains on GeoLife are aligned with the paper’s motivation that shared multi-resolution embeddings are especially useful when explicit POI identifiers are unavailable or fine-grained embeddings receive sparse updates. The reduction in embedding parameters is also directly quantified, and the ablation and training-paradigm comparisons provide additional support for the proposed design.

However, the evidence does not fully characterize two possible limitations. First, the experiments assume a single correct user for every trajectory and do not quantify how often identical or highly similar trajectory representations are associated with different users. Second, the aggregate results do not directly test whether sharing grid embeddings harms geographically close but behaviorally unrelated POIs. These omissions limit the characterization of when MiLES may fail, but they do not invalidate the main empirical conclusion that MiLES improves the evaluated TUL models under the studied online setting.

**Requested Changes:**

1. **[Strengthening; not critical for acceptance] Analyze ambiguity in the single-user TUL formulation.** Multiple users may generate identical or nearly identical spatial-temporal trajectory patterns. Although timestamp features may distinguish some cases, time-of-day may not be informative for every trajectory. Under the current single-label evaluation, if the same observable pattern occurs once with user (U_1) and later with user (U_2), predicting (U_1) both times is counted as correct for the first instance and incorrect for the second. In the online setting, these examples also produce conflicting sequential gradients, potentially leading to order-dependent predictions or apparent forgetting caused partly by label ambiguity. The authors should quantify how often similar trajectory patterns are associated with multiple users and report performance separately for low- and high-ambiguity examples. A top-(k), set-valued, or multi-label formulation could also be discussed. Gao et al. (2018) provide a relevant methodological precedent by formulating trajectory-based social-circle inference as multi-label prediction over users, although their labels represent the trajectory owner’s social contacts rather than multiple plausible trajectory generators.

 Q. Gao, G. Trajcevski, F. Zhou, K. Zhang, T. Zhong, and F. Zhang. “Trajectory-based Social Circle Inference.” *Proceedings of the 26th ACM SIGSPATIAL International Conference on Advances in Geographic Information Systems*, 2018.


2. **[Strengthening; not critical for acceptance]** Evaluate possible negative transfer caused by spatial sharing.
MiLES assumes that geographically proximate POIs benefit from shared representations across several grid levels. This assumption may fail in spatially heterogeneous regions where nearby venues have different functions and attract different user populations. In an online setting, an update from one POI immediately changes the shared embeddings used by nearby POIs, potentially causing negative transfer or order-dependent errors. The current concatenation and grid-resolution experiments provide indirect evidence about the specificity–sharing trade-off, but they do not directly evaluate this failure mode. The authors should compare MiLES with POI-only embeddings as a function of within-cell user-label or POI-category heterogeneity. This would clarify when multi-level spatial sharing is beneficial and when more adaptive or semantically informed sharing may be necessary.

---

> ### Author Response · Authors · 2026-07-05
>
> We thank the reviewer for the thoughtful feedback. Both points identify important boundary conditions for online TUL and for MiLES's spatial-sharing assumption, and we added two focused analyses to characterize them directly.
>
> **Ambiguity in the single-user TUL formulation**
>
> Ambiguity in the single-user TUL formulation is an important issue for TUL in general, not just our setting. If two users produce the same observable trajectory pattern, a single-label evaluation must mark only one as correct, even when several users are plausible from the input alone.
>
> To quantify this, we added an appendix analysis on Foursquare-NYC (Table 20). For each trajectory, we compute the longest common subsequence between its hour-precise (POI, hour) tuples and those of previous trajectories from different users, normalized by trajectory length, giving a stream-respecting measure of prior different-user ambiguity.
>
> Most trajectories have little or no prior different-user overlap, but a small high-overlap subset (~7%) remains and is much harder for both embeddings. On no-overlap and partial-overlap trajectories, BiTULER stays around 63 to 65% top-1 accuracy, with MiLES consistently ahead of the POI-only embedding. In the high-overlap bin, top-1 accuracy drops to roughly 24% for both methods, while top-5 stays substantially higher, around 39 to 40%. Even these are far above chance for an 800-user task (random top-1 is about 0.125%), so the models still recover substantial signal, and the true user is often among the most plausible candidates even when the top prediction is unreliable.
>
> These results support the reviewer's concern. Some errors reflect trajectories whose observable POI and hour sequence closely matches a previous one from another user. MiLES improves the no-overlap and partial-overlap bins, but high-overlap cases remain difficult for both methods.
>
> One way to mitigate this ambiguity is to add discriminative features, such as the day of the week or finer contextual signals, that separate otherwise identical POI-and-hour patterns. Even without them, top-k prediction is often a valid strategy in practice. In ride-sharing, for example, recommending the k most likely users to share a ride with keeps a model trained under the standard single-label formulation useful, which is reflected in the higher top-5 accuracy we observe. For such applications, however, set-valued or multi-label formulations could be a more faithful problem statement, following the precedent the reviewer notes for trajectory-based social-circle inference (Gao et al., 2018), which we now cite.
>
> Changes:
>
> - Added an appendix overlap analysis (hour-precise LCS over (POI, hour) tuples) reporting BiTULER by no, partial, and high prior different-user overlap.
> - Discussed overlap-reduction features (e.g. day of the week), top-k prediction (e.g. ride-sharing), and future set-valued or multi-label formulations, citing Gao et al. (2018).
>
> **Negative transfer from heterogeneous shared cells**
>
> Spatial sharing could in principle cause negative transfer when nearby POIs are behaviorally heterogeneous. To check this, we added a main-text experiment on Foursquare-NYC (Table 9). For each shared MiLES grid cell we computed normalized entropy over POI categories and user labels, assigned each trajectory its shared cells' average diversity, and split into low, medium, and high tertiles.
>
> MiLES improves all metrics across category-diversity bins, including the high-diversity bin. This suggests that category-diverse cells can still encode useful neighborhood-level information not captured by venue categories alone. For example, a shopping district can contain diverse POI categories while still serving a coherent local function. For user diversity, MiLES retains higher top-5 accuracy and macro F1 across all bins, while top-1 accuracy is essentially tied in the high-diversity bin (41.00% vs. 41.12% for POI-only).
>
> The reviewer's concern has merit, and the essentially tied top-1 in the high user-diversity bin is consistent with some negative transfer in exactly the cells the reviewer highlights. We are cautious about attributing it cleanly, because under our fixed total embedding dimension MiLES necessarily reduces POI-level capacity, and the new dimension budget sweep (Table 8) shows that this capacity reduction alone carries a substantial cost. The shared levels therefore appear at worst less helpful for exact top-1 identification in highly user-diverse cells, rather than clearly harmful. Either way, this is a useful boundary condition, and it motivates future adaptive sharing mechanisms, such as dynamically splitting diverse cells.
>
> Changes:
>
> - Added a main-text category- and user-diversity diagnostic (Table 9) reporting POI-only vs. MiLES by low, medium, and high diversity.
> - Connected the high user-diversity result to the fixed-capacity trade-off from the budget sweep, and added future-work discussion of adaptive sharing.

---

### Review · Reviewer_neBF · 2026-06-28

**Summary Of Contributions:**

## Summary of contributions

The paper studies Trajectory-User Linking (TUL) in an online / streaming setting, which appears to be genuinely underexplored. It contributes (i) a systematic evaluation of several state-of-the-art TUL backbones under a prequential (test-then-train) online protocol; (ii) MiLES, a drop-in spatial embedding layer that maps each check-in to multiple hexagonal grids of increasing cell size, assigns each level a decaying embedding dimension and a learnable scalar weight, and concatenates the levels; and (iii) an empirical study (gains, ablations, a simulated-drift experiment, a batch/retraining/online comparison) plus a destination-prediction generalization study. Code is provided.

Overall I see the paper as providing a useful new problem framin (online TUL) plus a broad empirical study, using a sensible but not-novel multi-scale embedding whose real contribution is the online instantiation.



## Strengths

- **Well-motivated, underexplored problem.** Framing TUL as online learning is sensible and, as far as I can tell, novel for this task.
- **Breadth of experiments.** Six backbones × three datasets (two user-count variants each) × three training paradigms, plus a separate destination-prediction task. I think this systematic benchmarking is relevant for TMLR.
- **Simple, modular, drop-in method** with concrete, directly-measured benefits (reduced parameter count, mostly reduced wall time).
- **Good ablation coverage** (levels, learnable weights, variable dimensionality, hexagon vs. square, concatenation vs. summation) and **reproducibility** (anonymized code, full hyperparameter grid, tuning protocol).
- **Honest limitations section** that pre-empts several concerns (delayed feedback, unbounded memory, limited drift coverage).

**Audience:**

Yes

**Audience Explanation:**

If my concerns are addressed, I think it would be of interest due to its use of multiscale modelling for spatial tasks and online learning, although the specific study is quite niche.

**Claims And Evidence:**

No

**Claims Explanation:**

# Full review

> **Note:** I wrote this offline without strictly following the TMLR form, so I have tagged each major item by priority:
> - **[must / should]**  the authors must address/refute this for me to recommend acceptance;
> - **[suggest]** optional.

## Major comments / questions / suggestions

### Claims and significance
- **[must] Fair presentation of the key gains.** The abstract claims "up to 24%", but I struggle to see where this comes from: it is not the typical gain across settings - it is the single best cell on GeoLife (the dataset with no POIs and the largest gains), not a representative result. It is also  inconsistent: the abstract states "up to 24%" while Section 6 reports the largest GeoLife gain as 8.72 percentage points (which I think is a 23% relative gain...?). The abstract and the paper must give a fair summary of the gains, e.g. report the *typical* range (roughly 1–3 points / ~2–5% relative on the Foursquare datasets).

- **[must] Use of "significant" / "significantly" (and "substantial") without any statistics.** The paper describes its central performance claims with the language of statistical significance while providing no test, no variance, and no effect size. The word also appears in its proper statistical sense for the ablation study (Wilcoxon, Table 7), so the two usages are confusing to me. The stronger-sounding claims are precisely the unsupported ones. I feel the main results (Table 2 / 13) must be backed by variance / confidence intervals and ideally paired significance tests extended to the Default-vs-MiLES comparison or the significance language must be corrected.

### Role of the level weights w_l (non-identifiability)

- **[must] The interpretation of w_l.** Because the embedding tables `Z_l` and the downstream projection `W` are both learnable and the levels are simply concatenated, `w_l` is **non-identifiable** from the forward map. Concretely, the model is invariant under the per-level rescaling (please correct me if I have this wrong.) Consequently, `w_l` adds no capacity per se, and the interpretation of `w_l` as the "relative importance of each spatial level" - and the Fig. 6 narrative about relying on coarser levels after drift - is not sensible as far as I can see. One cannot read level importance off a non-identifiable parameter, as far as I know.

- **[should] Test why effect of -WL makes a difference.** The `-WL` ablation does show an effect, but since the parameter is non-identifiable (I think), that effect must be an *optimization* one (an effective per-level learning rate / gradient scaling), not a representational one. Please reframe `w_l` accordingly and test it directly — e.g. compare against explicit per-level learning rates — and, if weight decay / L2 is used, disentangle the effect from a regularization-path effect (regularization breaks the rescaling symmetry).

### Grid-based and continuous embedding alternatives

- **[should] Compare against, or justify omitting, Hu et al. (2022).** The multi-scale idea is not new, as the related work notes. Hu et al. is the closest existing grid-based approach (also comes with fixed-size hash tables adressing the limitation in Section 9). Please either compare against it or justify the omission (e.g. a concrete reason it is unsuited to TUL) and explain why the proposed multi-scale embedding is better suited to TUL.

- **[should] The continuous-encoder comparison rests on a weak baseline.** The paper outscopes continuous encoders (Space2Vec / Sphere2Vec, and other uncited ones) by treating the Fourier basis as a stand-in but the Fourier baseline is weak (it underperforms even plain POI lookup), and Space2Vec is the *named* method whose principle MiLES claims to share. I am not insisting on a continuous baseline, but the authors should either (a) add a purpose-built location encoder (Space2Vec / Sphere2Vec, or a GeoCLIP-style hierarchical encoder) so the grid-vs-continuous question is actually tested, or (b) explain precisely why continuous encoders are unsuited to TUL. In either case be more careful with claims that overinterpret MiLES vs. continuous models on the basis of the Fourier comparison.

### Total dimensionality of the embedding

- **[should] Analyze the effect of total embedding dimension.** The paper emphasizes parameter/efficiency trade-offs, and Table 12 sweeps (512,1024)) for tuning, but it never *analyzes or reports* the effect of the total dimension. For a method whose central claim is how to *spend* an embedding budget, I would expect a budget sweep showing how MiLES vs. POI-lookup vs. a continuous encoder behave as `d` varies. Otherwise it is difficult to see if te ranking holds across budgets or is an effect of the single tuned dimension. This would also let the authors test the "gains are conservative because d was fixed" claim, which Table 8 currently appears to undercut.

### Presentation

- **[suggest] Readability / flow.** The paper is dense regarding teh figures/tables (15 tables, 13 figures), and large tables frequently interrupt the narrative. I think this is about organization rather than length. Perhaps consider moving the per-variant tables to the appendix (keeping averaged summaries in the main text), consolidating closely-related figures, and adding more signposting. This does not affect my assessment.

## Minor

- **Baseline embeddings.** Could the authors clarify exactly what the "Fourier" baseline uses (random Fourier features? which scales?), and provide a brief complexity / parameter comparison between the Fourier basis and the MiLES embeddings? (A spherical basis would be the geometrically principled continuous choice, though at single-city scale I would not expect it to matter much...?)
- **P2, "...extends the principle of multi-scale spatial representation":** a reference to the work being extended would help here (I appreciate it comes later, but a pointer at this point would aid the reader).
- **P2, "into multiple segments":** is it clear what a *segment* is at this point in the text?
- **P2:** what precisely is meant by "embedding" vs. "representation" in this context?
- **P3:** I do not follow the subscript notation for the `\mathbb` term with the "," — why is the "," included?
- **P6:** Algorithm 1 is quite simple — is it needed in the main paper, or could it move to an appendix?
- **Geolocalization literature.** For multi-scale spatial representation, I'd suggest also reviewing the image-geolocalization literature, where hierarchical / multi-scale grids are long established (PlaNet, Weyand et al. 2016; the hierarchical multi-partition model of Müller-Budack et al. 2018). In particular, GeoCLIP (Vivanco et al., NeurIPS 2023) proposes a hierarchical, multi-resolution location encoder with an exponential per-scale assignment that looks closely related to MiLES's learnable level weights and geometric dimension decay.
- *Which hyperparameters does each row of the online-vs-retraining comparison use?** Sec 6.1 states this comparison uses batch-tuned hyperparameters for both modes but the "Online" rows of Table 9 do not match the batch-tuned relative gains in Table 8 and instead match the online-tuned main experiment (Table 2). E.g. Table 9 NYC online 60.12 to 61.61 (+2.48%) vs. Table 8's batch-tuned +4.32%. Table 9 GeoLife online matches Table 2 to two decimals. Please clarify which configuration each row uses and fix any issues.
- **Parameter counts.** I tried to compute the reported MiLES parameter counts (e.g. 21.97M for NYC) from the stated hyperparameters. With 34,383 places, d=1024, alpha=2 and 2–3M lower regardless of level count.
## Trivial / nitpicking

- Please check punctuation around equations, e.g. (3).

**Requested Changes:**

Se my full review above.

---

> ### Author Response · Authors · 2026-07-05
>
> We thank the reviewer for the careful and constructive review, and in particular for the point on the non-identifiability of the level weights, which led us to a clearer and better-supported account of what they do. As the comments are extensive, we split our response across four comments, following the structure of the review.
>
> **Fair presentation of the key gains**
>
> We agree, and have rewritten the abstract, contributions, and results so they no longer lead with the single best number. The abstract and contributions now state relative top-1 gains of roughly 2 to 5% on the POI-rich Foursquare datasets, and describe GeoLife qualitatively, as the dataset where, lacking POI features, multi-level spatial embeddings are particularly beneficial, rather than highlighting a single maximum percentage.
>
> In the results section we now report the GeoLife gains as absolute maxima over the backbones, up to 8.72 pp. top-1 (BiTULER) and 6.98 pp. top-5, alongside the typical Foursquare range, and we no longer cite the previous ~24% relative figure. That number was a maximum over a different backbone (MainTUL).
>
> Changes:
>
> - Abstract, contributions, and results now report the Foursquare range (~2 to 5% relative), describe GeoLife qualitatively, and drop the ~24% headline. GeoLife gains appear as absolute maxima (8.72 pp. top-1, 6.98 pp. top-5).
>
> **Use of "significant" / "substantial" without statistics**
>
> We agree that using the language of statistical significance for the descriptive performance claims was confusing. We have gone through the manuscript and removed the statistical connotation from the descriptive claims, keeping "significant" only where it refers to an actual test.
>
> More importantly, we now support the Default-vs-MiLES comparison with the variance and paired significance tests the reviewer asked for. Table 18 reports the paired $\Delta$ MiLES effects, with a standard deviation of at most 0.78 pp. and improvements in all 18 model $\times$ dataset settings on every metric, and Table 19 adds a per-model one-sided Wilcoxon signed-rank test (N = 10 paired observations) that is significant for every model, dataset, and metric.
>
> Changes:
>
> - Reserved "significant" and "substantial" for test-backed claims throughout.
> - Added Table 18 (paired $\Delta$ MiLES standard deviation at most 0.78 pp., improving in all 18 settings) and Table 19 (per-model Wilcoxon test, N = 10, significant everywhere).
>
> **The interpretation of $w_l$ (non-identifiability)**
>
> The reviewer is correct. We were aware of the weights' non-identifiability and shared the reviewer's hypothesis that they act as per-level learning-rate modifiers, but failed to make this evident enough in the previous version, which we have now corrected.
> The forward map is invariant under the per-level rescaling (the scale can equally move into the backbone's input weights $W_l$), so a single $w_l$ adds no capacity and cannot be read as a level's importance. Since we use no weight decay, the symmetry is exact, so the effect cannot be a regularization-path artifact. The approach section now states the exact symmetry and that no weight decay is used, and we removed the "relative importance" reading and the drift narrative around Figure 6.
>
> To make the point concrete, we added a figure tracking the identifiable contribution magnitude $g_l = w_l |Z_l| |W_l|$. It shows that the POI level dominates throughout even though $w_3$ is the largest weight, confirming the raw weights are not directly interpretable as a level-importance score. Figure 6 is now purely descriptive of the optimization dynamics.
>
> Changes:
>
> - The approach section now states the exact rescaling symmetry and that no weight decay is used, concluding $w_l$ adds no capacity and is not a level-importance score, and we removed the "relative importance" and drift narrative from Figure 6.
> - Added a contribution-magnitude figure ($g_l = w_l |Z_l| |W_l|$) where the POI level dominates throughout, so raw $w_l$ cannot be read as importance.

---

> > ### Author Response · Authors · 2026-07-05
> >
> > **Why the learnable level weights make a difference**
> >
> > We agree that since $w_l$ is non-identifiable, its effect must act through the optimization rather than through added capacity. We now frame $w_l$ as a per-level forward scale, the magnitude at which each level enters the backbone, and separate this from the learning-rate reading.
> >
> > First, a new component decomposition shows the per-level re-weighting is carried almost entirely by $w_l$. The embedding and input-weight norms barely vary across levels or between MiLES and -WL. Matching that re-weighting without $w_l$ would require coordinated changes across the whole input block reading each level, which -WL never manages, as it stays near its initialization, while with the weights present the per-level magnitude shifts substantially over the stream. The single scalar weights appear to be better conditioned as a control over each level's forward scale than the high-dimensional input weights, which is why -WL leaves a small but consistent gap. In our view this ease of adjusting the entry magnitude online is the main benefit of the level weights.
> >
> > Second, we tested the reviewer's suggestion that $w_l$ acts as an effective per-level learning rate. Replaying each seed's recorded $w_l(t)$ as a per-step learning-rate multiplier on the embedding tables (starting from -WL, Table 24) recovers only a small fraction of the performance gained with the learnable weights.
> >
> > Changes:
> >
> > - Reframed $w_l$ as a per-level forward scale acting through the optimization, not a representational parameter.
> > - Added a component-decomposition figure showing the re-weighting is carried by $w_l$ (embedding and input-weight norms barely vary).
> > - Added an appendix experiment (Table 24) replaying $w_l(t)$ as explicit per-level learning rates, which does not reproduce the effect, ruling out the effective-learning-rate reading.
> >
> > **Comparison with Hu et al. (2022)**
> >
> > We appreciate this suggestion and now include a DeeprETA-style embedding as a baseline (Table 6). We adapt it to the online setting as a multi-resolution square-grid concatenation with uniform per-level dimensions and no learnable weights, essentially MiLES with its variable dimensions (-VD) and level weights (-WL) removed and a square grid, tuned on the same protocol as every other method, with POI kept as the finest level. We omit only its feature-hashing step, which bounds table size and is an orthogonal, memory-side mechanism, compatible with MiLES's tables, and we now note it in the related work and limitations.
> >
> > MiLES outperforms this baseline on both Foursquare datasets across all six backbones (e.g. NYC 58.39 vs 56.55 top-1). On POI-free GeoLife the two are close, with the DeeprETA-style grid slightly ahead in aggregate (44.43 vs 43.93). This reflects our tuning protocol, which emulates the online setting where no data from the target stream or distribution is available for tuning in advance. Each method is tuned once on Foursquare-TKY and its grid is transferred unchanged, so DeeprETA's grid happens to suit GeoLife's larger extent better than MiLES's does. Holding the grid fixed (Appendix, B.3) removes this advantage. MiLES then wins on GeoLife across all six backbones, confirming the small aggregate gap reflects the transferred grid rather than the embedding design, consistent with the -WL/-VD ablations.
> >
> > Changes:
> >
> > - Added a DeeprETA-style embedding baseline (Table 6), tuned on the same protocol, with feature hashing omitted as an orthogonal memory-side mechanism.
> > - MiLES wins on both Foursquare datasets across all backbones and is competitive on GeoLife, and a matched-grid comparison (Appendix B.3) shows its embedding design wins on GeoLife too once the grid is fixed.

---

> > > ### Author Response · Authors · 2026-07-05
> > >
> > > **The continuous-encoder comparison**
> > >
> > > We thank the reviewer for this point, which identified a genuine error in our results. The Fourier rows in Tables 2 and 13 were run with an untuned frequency scale instead of the tuned value. That value (Fourier feature scale 4000) is listed in our hyperparameter table, but we failed to apply it to these runs. We have re-run the Fourier baseline at the tuned scale across all six backbones and three datasets and updated both tables.
> > > With this correction the Fourier encoder is a strong baseline. It is equivalent to the grid variant of the Space2Vec encoder up to the tuned wavelength range. The one substantive difference from Space2Vec and GeoCLIP is that they pre-train their encoders on the full dataset, which our single-pass online setting does not permit. On the POI-rich Foursquare datasets MiLES still outperforms the Fourier encoder across all backbones and metrics. On the POI-free GeoLife dataset it is now the strongest embedding, outperforming MiLES by roughly 10 percentage points top-1 on average (7 pp. top-5). We also add the Theory variant of Space2Vec as an explicit baseline, our Fourier baseline being its Grid variant. It stays within roughly 1.5 pp. of the Fourier encoder on every dataset and gives the same result.
> > >
> > > We believe this advantage reflects how location is represented on each dataset. On the Foursquare datasets the finest embedding level indexes individual POIs, giving each venue a distinct representation. GeoLife has no POIs, so nearby points fall in the same grid cell and receive an identical embedding even at high resolution, whereas a continuous encoder assigns each point a distinct code. This finer spatial detail appears more valuable on GeoLife's dense, continuous trajectories.
> > >
> > > Changes:
> > >
> > > - Fixed an error where the Fourier rows in Tables 2 and 13, and the Fourier rows of the destination-prediction experiments, used an untuned frequency scale rather than the tuned value from our hyperparameter table, and re-ran the baseline at the tuned scale across all backbones and datasets. The destination-prediction results are qualitatively unchanged.
> > > - With the correction MiLES leads the Fourier encoder on both Foursquare datasets across all backbones, while on GeoLife the continuous encoder is now the strongest embedding. We removed the claim that MiLES outperforms all embeddings on all datasets, and the contributions and conclusion now present MiLES as an effective alternative to the standard POI-only lookup embedding, with a continuous encoder more effective on denser GPS data such as GeoLife.
> > > - Added the Theory variant of Space2Vec as an explicit baseline and noted that our Fourier baseline is its Grid variant up to the tuned wavelength range.
> > > - Added a hypothesis for the difference. On POI data the finest level identifies each venue precisely, while on POI-free GeoLife the grid cannot resolve nearby points, so a continuous encoder that keeps fine spatial detail does better. The continuous-target destination-prediction task is consistent with this.
> > >
> > > **Effect of the total embedding dimension**
> > >
> > > We added the budget sweep the reviewer asked for (Table 8), varying the total location embedding dimension from 128 to 2048 with the hidden size fixed. MiLES leads at every budget and metric, most clearly when the budget is tight (8.6 pp. at $d = 128$), so the ranking is not an artifact of the single tuned dimension.
> > >
> > > We also make the parameter view explicit. At $d = 1024$ MiLES matches the POI lookup's $d = 2048$ accuracy (61.60 vs 61.69 top-1) using under a third of its embedding parameters, so its gains do not come from extra capacity. We keep $d = 1024$ for the main experiments as the largest dimension in our tuning grid, with diminishing returns beyond it.
> > >
> > > Changes:
> > >
> > > - Added Table 8, a total-dimension budget sweep (128 to 2048, hidden size fixed) where MiLES leads at every budget, and noted MiLES at $d = 1024$ matches the POI lookup at $d = 2048$ using under a third of the parameters.
> > >
> > > **Parameter counts**
> > >
> > > We added a full per-level parameter breakdown (Table 16) reproducing the reported totals (e.g. 21.97M on Foursquare-NYC). At $d = 1024$ the geometric schedule gives the POI level 547 dimensions and the three coarse levels 273, 136, and 68. The POI table dominates while the coarse tables stay small, since the grids hold far fewer cells than there are POIs. This should reconcile the discrepancy the reviewer found.
> > >
> > > Changes:
> > >
> > > - Added Table 16 with the per-level dimension and parameter breakdown reproducing the reported totals.

---

> > > > ### Author Response · Authors · 2026-07-05
> > > >
> > > > **Geolocalization literature**
> > > >
> > > > We added the suggested references. PlaNet (Weyand et al., 2016) and Müller-Budack et al. (2018) are discrete cell-classification methods, and GeoCLIP (Vivanco et al., 2023) is a continuous location encoder. GeoCLIP uses a different frequency band per level, which parallels MiLES. However, each level of GeoCLIP spans the full embedding and the levels are summed with equal weight, whereas MiLES gives each level its own slice of the embedding dimensions through the geometric decay and learns how much each level contributes. We now note this resemblance at the multi-resolution level in the related work.
> > > >
> > > > Changes:
> > > >
> > > > - Added PlaNet, Müller-Budack et al., and GeoCLIP to the related work, noting GeoCLIP's multi-resolution structure resembles MiLES while its per-level mechanism differs.
> > > >
> > > > **Online-vs-retraining configuration (Table 11)**
> > > >
> > > > Thank you for catching this. Every row of the retraining comparison now uses the batch-tuned configuration, including the online rows, so their relative gains match batch-tuned Table 10 exactly (e.g. Foursquare-NYC +4.32% top-1). We added a sentence noting these differ slightly from the online-tuned main results for this reason, so the two are not confused.
> > > >
> > > > Changes:
> > > >
> > > > - Corrected the retraining table so every row, including the online rows, uses the batch-tuned configuration (consistent with Table 10), and noted these differ from the online-tuned main results.
> > > >
> > > > **Remaining minor points**
> > > >
> > > > We addressed the remaining minor items in the revised manuscript. We added a citation pointer to pre-existing multi-scale work at its first mention, reworded the ambiguous "segments" to "parts", explained the comma in the $Z_{l,h_l}$ subscript (which separates the level index from the within-level row index), moved Algorithm 1 to the appendix, and checked the punctuation around the equations. We also agree with the broader point that some large floats currently interrupt the narrative, and we will make a final pass over figure and table placement, consolidation, and signposting once the content of the revision has settled, so the layout is not disrupted by further edits.